# Understanding and Mitigating Numerical Sources of Nondeterminism in LLM Inference

**Jiayi Yuan**[1]* **Hao Li**[2]* **Xinheng Ding**[2] **Wenya Xie**[2] **Yu-Jhe Li**[3]
**Wentian Zhao**[3] **Kun Wan**[3] **Jing Shi**[3] **Xia Hu**[1] **Zirui Liu**[2]
[1]Rice University [2]University of Minnesota Twin Cities [3]Adobe Inc.
{jy101,xia.hu}@rice.edu
{li003703,ding0499,xie00470,zrliu}@umn.edu
{jhel,wezhao,kuwan,jingshi}@adobe.com

## Abstract

Large Language Models (LLMs) are now integral across various domains and have demonstrated impressive performance. Progress, however, rests on the premise that benchmark scores are both accurate and reproducible. We demonstrate that the reproducibility of LLM performance is fragile: changing system configuration, such as evaluation batch size, GPU count, and GPU version, can introduce significant differences in the generated responses. This issue is especially pronounced in reasoning models, where minor rounding differences in early tokens can cascade into divergent chains of thought, ultimately affecting accuracy. For instance, under bfloat16 precision with greedy decoding, a reasoning model like DeepSeek-R1-Distill-Qwen-7B can exhibit up to **9% variation in accuracy and 9,000 tokens difference in response length due to differences in GPU count, type, and evaluation batch size.** We trace the root cause of this variability to the non-associative nature of floating-point arithmetic under limited numerical precision. This work presents the first systematic investigation into how numerical precision affects reproducibility in LLM inference. Through carefully controlled experiments across various hardware, software, and precision settings, we quantify when and how model outputs diverge. Our analysis reveals that floating-point precision—while critical for reproducibility—is often neglected in evaluation practices. Inspired by this, we develop a lightweight inference pipeline, dubbed *LayerCast*, that stores weights in 16-bit precision but performs all computations in FP32, balancing memory efficiency with numerical stability. Code is available at https://github.com/nanomaoli/llm_reproducibility.

## 1 Introduction

Large Language Models (LLMs) are increasingly being deployed in everyday scenarios, powering applications from chatbots [2] to automated coding tools [25] and personalized healthcare agents [6]. As their impact grows, rigorous benchmarking and evaluation become critical to measure real progress and ensure reliability, safety, and fairness [7, 37]. There are two commonly used evaluation strategies. The first one is to use *greedy decoding* by setting the temperature to zero, which produces deterministic outputs, and reports the result from a single run. The second uses *random sampling* with a non-zero temperature and reports performance using the Pass@K metric [8], i.e., the probability that at least one of K independent solution attempts will succeed.

In this paper, we highlight a commonly overlooked factor in both evaluation settings: **numerical precision**. First, for the greedy decoding setting, this factor undermines the assumption of determin-

---

*Equal contribution

39th Conference on Neural Information Processing Systems (NeurIPS 2025).

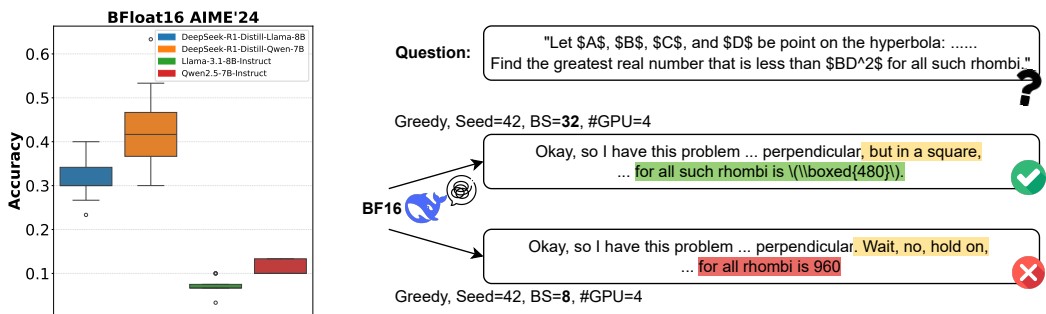

Figure 1: **Left:** Under BF16 precision and greedy decoding, the model's output can vary significantly depending on factors such as GPU count, evaluation batch size, and GPU hardware version. **Right:** For example, changes in evaluation batch size alone can lead to noticeable differences in responses, which is often ignored and not standardized by evaluation benchmarks.

ism–even with the same prompt and random seed, the generated output can still differ significantly between different hardware and system configurations. Second, for the random sampling setting, the numerical error demands a larger number of runs to control the variance. Surprisingly, we found that the results can be significantly different under the greedy decoding when changing the evaluation batch size, number of GPUs, or GPU versions. Through analysis, we found that the root cause of this problem is the non-associative property floating-point arithmetic, meaning $(a + b) + c \neq a + (b + c)$ due to finite precision and rounding errors. This issue is particularly problematic for recent reasoning-focused models [13], which generate very long chains of thought. In such cases, small numerical errors accumulate during the token generation process, eventually leading to significant differences in output across different runs. As illustrated in Figure 1, the model produces significantly different outputs when the number of GPUs, evaluation batch size, or hardware versions change—even if the same random seed and greedy decoding are used. This inconsistency makes it difficult to reproduce results, posing a serious problem for measuring the progress.

We highlight the significance of this issue for several critical reasons. First, many researchers report benchmark performance based on a single inference run with a fixed random seed to reduce computational cost. However, as shown in Figure 1. This practice can lead to misleading conclusions about model performance. Second, even when performing random sampling with multiple independent runs, the averaged results can still vary significantly due to hardware and system-level nondeterminism. Moreover, when researchers report standard deviations without accounting for this numerical nondeterminism, they risk severely overestimating a model's true uncertainty, since the reported variance reflects a mixture of intrinsic model uncertainty and variance introduced by finite numerical precision. When results cannot be exactly reproduced, it becomes difficult to distinguish whether improvements are from better methods or merely random variation.

To fill this gap, we conduct a comprehensive analysis of how numerical precision and hardware configurations affect the reproducibility of LLMs. **Our findings show that inference using the commonly adopted BF16 precision is highly sensitive to variations in hardware and system configurations, such as tensor parallel size, evaluation batch size, and GPU types.** These hardware-related factors are often beyond users' control and can vary significantly due to resource availability, yet they are often overlooked in current LLM evaluation methods. We observe that increasing the number of mantissa bits in the numerical format—such as using FP16 or FP32—can significantly mitigate this issue. Based on these findings, we propose an optimized inference pipeline that performs all computations in FP32 while retaining model weights in BF16 precision. This approach effectively balances memory efficiency and reproducibility. Specifically, our contributions and suggestions can be summarized as follows:

- **Extensive analysis of how numerical precision affects reproducibility in both greedy decoding and random sampling scenarios.** Our finding suggests that due to the limited precision, the model performance is highly sensitive to variations in hardware and system configurations, such as tensor parallel size, evaluation batch size, and GPU types.
- **Practical suggestions for reproducible reasoning to the community**. Based on our findings, we suggest (1) If sufficient computational resources are available, please use random sampling (non-zero temperature) with multiple runs. Report the mean accuracy, average answer length, and

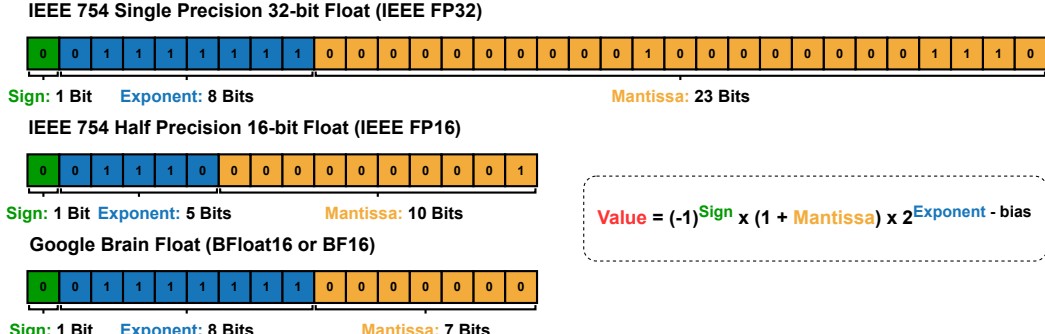

Figure 2: Floating-point format of FP32, FP16 and BF16.

error bars. Note that with 16-bit precision and small datasets, more trials are needed for stable results in general. (2) If using greedy decoding with a single run, please use FP32 precision to improve the reproducibility of your results.

- **An optimized FP32 inference pipeline**. We propose an optimized inference pipeline called LayerCast, which performs all computations in FP32 while retaining model weights in BF16 precision. This approach effectively balances memory efficiency and reproducibility. We release it as a patch to vLLM and can be used with just a few lines of code change.

## 2 Preliminary and Analysis

### 2.1 Current Practices on LLM Reproducibility

There are two widely adopted evaluation strategies. The first one is *greedy decoding*, where temperature is set to zero and model always select the token with highest probability. The second strategy employs *random sampling* with a non-zero temperature, and evaluates performance using the Pass@K metric [8]. Below, we introduce the commonly adopted experimental setting for the reproducibility.

**Greedy decoding** is a deterministic text generation strategy where the model always selects the token with the highest probability at each step. In theory, this approach should produce identical outputs given the same input and model parameters. However, in this paper, we show that even greedy decoding can yield different results across runs due to numerical precision issues.

**Random sampling** selects output tokens based on the model's probability distribution with a non-zero temperature to introduce randomness. The most commonly used evaluation metric is the mean accuracy across multiple independent trials, which is equivalent to Pass@1 [8].

**Deterministic libraries and Random Seeds**. Random seeds in LLM generation control the pseudo-random number generator that selects tokens when using non-greedy decoding. It is a standard practice to fix random seeds; however, this is not always sufficient for ensuring full determinism in greedy decoding. Even with a fixed seed, differences in computation order can alter the sequence of the pseudo-random number generator [16]. Meanwhile, frameworks like PyTorch and TensorFlow provide flags for deterministic behavior (e.g., `torch.use_deterministic_algorithms(True)` [2]). These ensure that certain operations avoid algorithms that could produce different results across runs. However, in practice, it can still produce nondeterministic results even with these flags.

Despite these efforts, achieving perfect reproducibility in LLM inference remains a challenge. The next subsection delves into a crucial factor that often leads to nondeterministic behavior even with greedy decoding and the aforementioned practices in place.

### 2.2 Numerical Precision, Rounding Errors, and GPU Kernels

**Numerical precision** is critical for reproducibility [26]. Higher precision numbers have less rounding error, which can reduce variability in results. Examples include using FP32 for certain parts of computation—like softmax or attention scores—even if the model weights are in 16-bit precision.

---

[2]`https://pytorch.org/docs/stable/generated/torch.use_deterministic_algorithms.html`

Table 2: Two illustrative cases demonstrate how rounding error, together with the non-associativity of floating-point addition, can affect numerical results. Example 1 reveals accumulation error at both precisions; Example 2 exhibits a discrepancy only in BF16, while FP32 delivers identical results, illustrating higher-precision numeric types are more tolerant of rounding errors.

| Example | Sum Order | FP32 | BF16 |
|---------|-----------|------|------|
| $a, b, c = 0.1, -0.1, 0.2$ | $a + b + c$ | 0011111001001100110011001100**01** | 00111110010011**01** |
| | $a + c + b$ | 001111100100110011001100110011**10** | 00111110010011**10** |
| $a, b, c = 0.0016, 0.0027, 1.0$ | $a + b + c$ | 0011111110000000100011001110011 | 0011111110000000**1** |
| | $a + c + b$ | 0011111110000000100011001110011 | 0011111110000000**0** |

This practice can mitigate run-to-run differences, but it doesn't provide theoretical guarantees since floating-point arithmetic cannot represent all real numbers exactly. Even with the same precision, different hardware implementations or computation orders can lead to slightly different results. In this paper, we primarily focus on the importance of numerical precision and provide a detailed analysis of its impact on LLM inference reproducibility.

Table 1: Rounding error when the same true value (1.00012) is represented in three different numeric formats.

| Precision | Decimal | Rounding Error |
|-----------|---------|----------------|
| FP32 | 1.00012004375457761 | $\approx +4.38\mathrm{e}{-8}$ |
| FP16 | 1.0 | $\approx -0.00012$ |
| BF16 | 1.0 | $\approx -0.00012$ |

Figure 2 shows the format of BF16, FP16 and FP32. Table 1 illustrates the rounding error that occurs when the same true value is represented in three different numeric formats. As expected, lower precision formats like FP16 and BF16 introduce larger rounding errors compared to FP32.

Moreover, a key aspect of floating-point arithmetic that contributes to nondeterminism is its non-associativity. This means that the order in which numbers are added can affect the final result due to accumulated rounding errors. Table 2 provides illustrative examples of how the order of summation can lead to different results in FP32 and BF16. As the examples demonstrate, although rounding follows deterministic rules, non-associativity introduces nondeterminism, which is further amplified by the larger rounding error of BF16. This becomes particularly relevant in the parallel computations performed in GPUs during LLM inference.

In the context of LLMs, even small numerical variations in the logit values can affect the final token selection when the top probabilities are close. Floating-point arithmetic in GPUs exhibits non-associativity, meaning $(a + b) + c \neq a + (b + c)$ due to finite precision and rounding errors. This property directly impacts the computation of attention scores and logits in the transformer architecture, where parallel operations across multiple threads can yield different results based on execution order. Formally, for a reduction operation $\oplus$ over a set of values $\{v_1, v_2, ..., v_n\}$ (such as summing attention scores or computing softmax denominators), the result depends on the specific ordering of operations: $\bigoplus_{i=1}^{n} v_i = v_{\pi(1)} \oplus v_{\pi(2)} \oplus ... \oplus v_{\pi(n)}$, where $\pi$ is a permutation of indices $\{1, 2, ..., n\}$ determined by thread scheduling.

**GPU Kernels add numbers in different orders when changing system configuration.** In serving systems, several factors can change the order in which floating-point operations are executed, including: *(1) continuous batching* [36], which dynamically modifies the set of requests within a batch; *(2) different operator implementations*, such as Split-K versus Non-Split-K MatMul [23]; *(3) operator hyperparameters*, like the block size in MatMul or FlashAttention; *(4) collective operations in parallel settings*, such as AllReduce; and *(5) parallelization strategies*, such as tensor parallelism (TP), which distribute computation across multiple GPUs. Together, all these factors can affect the determinism of LLM inference. To address this issue, *Batch-invariant operations* [14], including batch-invariant FlexAttention, RMSNorm, and MatMul kernels, are introduced. These kernels guarantee that inference results remain deterministic regardless of batch size variations. Specifically, this approach achieves determinism by parallelizing computation along the batch dimension, thereby decoupling batch size from the final outputs of individual requests. However, as the name "batch-invariant" suggests, the technique is currently limited to handling variations related only to the batch dimension, making it robust to continuous batching and other batch-size–related changes, but not to other forms of nondeterminism like changing the TP sizes or GPU types.

# 3 Reproducibility Issues with Limited Numerical Precision

## 3.1 Experiment Setup

We conduct experiments on four recent LLMs, including two reasoning models: DeepSeek-R1-Distill-Qwen-7B, DeepSeek-R1-Distill-Llama-8B and two non-reasoning models: Qwen2.5-7B-Instruct and Llama-3.1-8B-Instruct [13, 34, 22], across five commonly used LLM evaluation benchmarks: AIME'24, MATH500, LiveCodeBench-Easy, LiveCodeBench-Medium, and LiveCodeBench-Hard [1, 15, 18]. To verify the generalizability of our findings, we also conduct additional experiments on a larger model (Qwen3-32B) and a diverse reasoning benchmark (GPQA Diamond) covering graduate-level science questions, with results provided in the Appendix. For reasoning models, we set the maximum output token length to 32,768 and for non-reasoning models, we set it to 2,048. We primarily use vLLM [20] as the inference backend; to verify that our findings are not backend-specific, we also conduct verification experiments using HuggingFace Transformers (see Appendix for details). In the random sampling experiments, we set *temperature* to 0.7 and *top-p* to 0.95. Our Evaluation implementation and prompt setting is adapted from SkyThought-evals [29], more details can be found in Appendix C.

For each model-task pair, we evaluate under 12 different runtime configurations, representing all combinations of 2 GPU types (NVIDIA L40S and A100), 2 GPU counts (2 and 4), and 3 batch sizes (8, 16, and 32), i.e., $2 \times 2 \times 3 = 12$ different configurations, to simulate the diversity of deployment environments encountered in real-world evaluations. Unlike decoding parameters or random seeds, these hardware-related factors are often beyond users' control and can vary significantly due to resource availability, yet they are often overlooked in current LLM evaluation methods.

To comprehensively quantify output instability and better analyze the impact of numerical precision on inference variability, we evaluate the reproducibility under both greedy decoding and random sampling settings. Specifically, for greedy decoding, we analyze:

- **Std@Acc** (Standard deviation of accuracy): For each numerical precision, we evaluate accuracy under the 12 different runtime configurations and compute their sample standard deviations, which serve as indicators of the stability of LLM inference outputs during greedy inference.
- **Avg_Std@Output_Length** (Average standard deviation of output length): We measure the length of output tokens, compute the sample standard deviation per example across 12 runtime configurations, and report the mean of standard deviations over the entire dataset. This provides an alternative perspective on the stability of LLM inference outputs during greedy inference.
- **Div_Index** (Divergence index): For the same question, ixf two or more responses produce identical token sequences up to a certain position, but generate different tokens after that position, we define the index of that position as the divergence index. A higher **Div_Index** indicates greater consistency across responses under different runtime configurations.
- **Avg_Std@top1_prob** (Average standard deviation of top-1 token prediction probability): Before divergence, all responses across different runtime settings produce identical top-1 tokens at every position. However, due to floating-point computation errors, the predicted probabilities of these tokens may vary across settings. To quantify this, we compute the standard deviation of the predicted probability for the top-1 token at each position across settings, then average over all positions from 0 to Div_Index and over all examples in a dataset. We define this metric as the Average Standard Deviation of Top-1 Token Prediction Probability, which serves as an indicator of the magnitude of numerical variation introduced by floating-point errors.

For random sampling, we analyze:

- **Pass@1**: For each numerical precision, we evaluate performance by running the model multiple times and computing the average accuracy accuracy (commonly referred to as Pass@1). Specifically, we use both 16 and 64 independent runs to compute mean accuracy for AIME'24, and 4 independent runs for MATH500. To assess the stability of random sampling-based decoding, we also report the standard deviation of Pass@1 across 6 runtime configurations, varying GPU count (2 vs. 4) and numerical precision (BF16, FP16, FP32).

## 3.2 Greedy Decoding $\neq$ Deterministic Output

**Contrary to common belief, our experiments reveal that greedy decoding does not guarantee deterministic outputs across different hardware and system configurations. However, using**

Table 3: Std@Acc of greedy decoding across 12 different settings (GPU types, GPU counts, and batch sizes) under BF16, FP16, and FP32 Numerical Precisions. Reasoning models also exhibit larger variance compared to non-reasoning counterparts. More results can be found in Appendix E.

| | AIME'24 | | | MATH500 | | | LiveCodeBench-Easy | | |
|---|---|---|---|---|---|---|---|---|---|
| | **BF16** | **FP16** | **FP32** | **BF16** | **FP16** | **FP32** | **BF16** | **FP16** | **FP32** |
| **DeepSeek-R1-Distill-Qwen-7B** | 9.15% | 5.74% | 0 | 1.04% | 1.12% | 0.12% | 1.67% | 1.28% | 0.37% |
| **DeepSeek-R1-Distill-Llama-8B** | 4.60% | 6.00% | 5.8e-17 | 1.59% | 0.73% | 0.23% | 2.31% | 1.92% | 0.29% |
| **Qwen2.5-7B-Instruct** | 1.71% | 1.45e-17 | 1.45e-17 | 0.83% | 0.36% | 1.16e-16 | 0.79% | 0.48% | 1.16e-16 |
| **Llama-3.1-8B-Instruct** | 1.92% | 1.30% | 0 | 0.94% | 0.34% | 0.13% | 1.00% | 0.67% | 0.25% |

Table 4: Standard deviation of output length of greedy decoding across 12 different settings (GPU types, GPU counts, and batch sizes) under BF16, FP16, and FP32 numerical precisions. The output length of reasoning models exhibit large variance. More results can be found in Appendix E.

| | AIME'24 | | | MATH500 | | | LiveCodeBench-Easy | | |
|---|---|---|---|---|---|---|---|---|---|
| | **BF16** | **FP16** | **FP32** | **BF16** | **FP16** | **FP32** | **BF16** | **FP16** | **FP32** |
| **DeepSeek-R1-Distill-Qwen-7B** | 9189.53 | 5990.32 | 0 | 2774.28 | 2090.46 | 138.75 | 5507.52 | 4282.78 | 262.55 |
| **DeepSeek-R1-Distill-Llama-8B** | 9348.59 | 7822.43 | 0 | 4015.00 | 2518.38 | 146.03 | 4732.85 | 3652.16 | 105.85 |
| **Qwen2.5-7B-Instruct** | 211.47 | 48.14 | 0 | 52.61 | 15.37 | 0 | 7.79 | 0.71 | 0 |
| **Llama-3.1-8B-Instruct** | 119.21 | 49.73 | 0 | 124.43 | 40.57 | 2.76 | 31.03 | 4.70 | 0.49 |

**FP32 precision helps a lot.** Table 3 presents the standard deviation of accuracy (Std@Acc) across 12 runtime configurations for BF16, FP16 and FP32 precisions. The results demonstrate a clear pattern: FP32 consistently achieves near-perfect reproducibility with negligible variance, FP16 shows moderate variability, while BF16 exhibits substantial instability. This pattern is particularly pronounced for reasoning models like DeepSeek-R1-Distill-Qwen-7B, where BF16 precision introduces up to 9% standard deviation in accuracy on AIME'24, compared to virtually zero variance under FP32. This finding is concerning because it suggests that **using different GPU types or varying the number of GPUs can *prevent you from reproducing* others' results, even when using greedy decoding.** Beyond accuracy variance, our results in Table 4 reveal that **BF16 precision also causes response lengths to vary significantly across different settings.** This raises significant concerns for recent work in efficient reasoning and long-to-short research [27], which needs to report response length metrics. Specifically, changing the system configuration can lead to a difference of up to 9,000 tokens in response length. This variability undermines the effectiveness of many existing efficient reasoning approaches. However, our findings show that using FP16 helps reduce output length variance to some extent, while FP32 offers the most consistent and reliable control over this variability. **Thus, we suggest if using greedy decoding, please use FP32 precision to improve the reproducibility of the efficient reasoning research.**

The performance gap clearly indicates that across different runs, the model generates different tokens. To investigate why this happens, we compared outputs from different runs. Since we are using greedy decoding, the top-1 probability token differs between runs at the point of divergence. Figure 3 shows an example of token logits when two runs diverge, illustrating how numerical precision errors can flip the order of the top-1 and top-2 token probabilities. Figure 3 also shows the histogram of probability differences between top-1 and top-2 tokens under FP32 precision. We observe that **for reasoning models, the token probability differences between the top two competing tokens are often minimal**.

To further understand how this observation interacts with the numerical errors, Figure 4 further illustrates the average standard deviation of the probability of prediction of the top-1 token (Avg_Std@Top1_Prob) at different numerical precision levels. BF16 shows significantly higher variance in top-1 token probabilities compared to FP16 and FP32. This increased variance arises because BF16's limited mantissa bits (7 bits compared to FP16's 10 bits) introduce larger errors in probability computations, increasing the likelihood of token flips when these fluctuations overlap with the small gap between top-1 and top-2 candidate tokens. In contrast, FP32's higher precision (23 mantissa bits) makes runtime variations nearly negligible. Together, these results demonstrate that **token selection during greedy decoding is highly sensitive to even small numerical variations because of the minimal probability differences between competing tokens.**

| Answer 1 | | Answer 2 | |
|---|---|---|---|
| Token | Prob. | Token | Prob. |
| know | 49.75% | have | 46.65% |
| have | 43.91% | know | 46.64% |
| need | 3.18% | need | 3.39% |
| 'm | 2.47% | 'm | 2.63% |
| 've | 0.49% | 've | 0.52% |
| ... | ... | ... | ... |

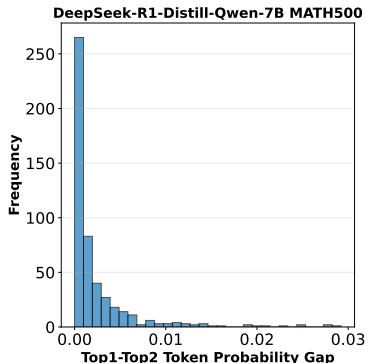

Figure 3: **Left:** the top-5 tokens and their predicted probabilities at the divergence index for two different answers to the same question in BF16. **Right:** The gap between the top-two competing tokens probability. We observe the token probability gap are often minimal in reasoning models.

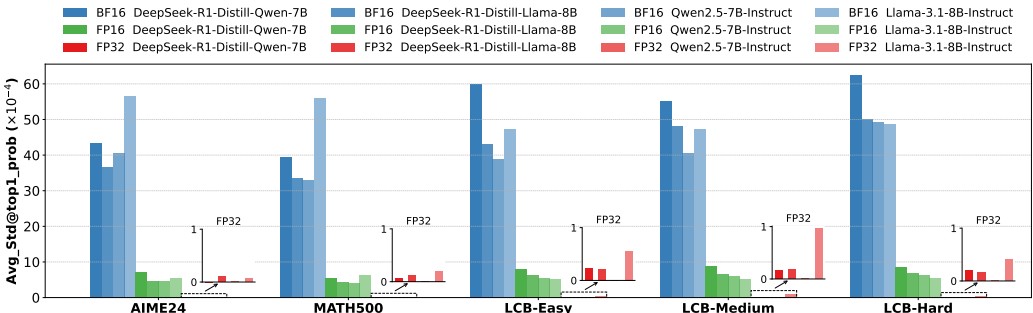

Figure 4: Avg_Std@top1_prob across 12 different settings for 4 models and 5 tasks, under BF16, FP16 and FP32. FP16 shows significantly lower variance compared to BF16. FP32 yields near-zero variance, demonstrating strong robustness to floating-point rounding errors.

The impact of numerical precision on output stability is further evidenced by divergence patterns in greedy decoding. Figure 5 shows the distribution of divergence points (Div_Index) across precision formats for DeepSeek-R1-Distill-Qwen-7B on MATH500. As precision increases from BF16 to FP32, we observe both fewer divergent examples overall and a significant shift in when divergence occurs. With FP32, almost all examples produce identical outputs across configurations, resulting in only 2.2% of the samples diverging. In contrast, with BF16, divergence frequently occurs early in generation despite the deterministic setting, with over 90% of examples showing divergence. When divergence happens in higher precision formats, it typically occurs much later in the sequence, limiting its impact on the final output and answer accuracy.

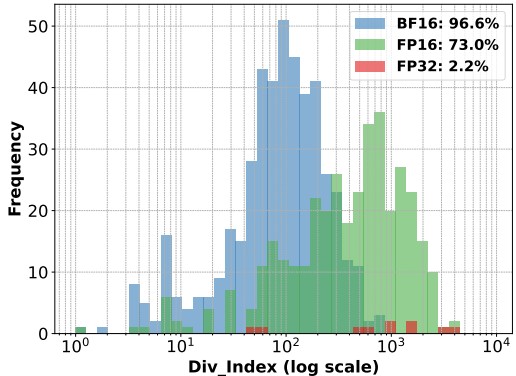

Figure 5: Distribution of Div_Index for DeepSeek-R1-Distill-Qwen-7B on MATH500 under BF16, FP16, and FP32. Higher numerical precisions lead to fewer divergent examples and a shift of divergence point to later token positions.

These results conclusively demonstrate that numerical precision is a critical factor in achieving truly deterministic outputs with greedy decoding, with higher precision formats providing substantially better reproducibility.

### 3.3 Random Sampling Has Reproducibility Issue, Too

One might argue that while greedy decoding is vulnerable to token-level instability from numerical precision, random sampling might be less sensitive due to its intrinsic stochasticity. **However, our experiments reveal that numerical precision significantly affects the stability and reproducibility**

Table 5: Standard deviation of Pass@1 performance (%) under different GPU counts and precisions. We emphasize that the reported values reflect **the variability of Pass@1 performance across 6 different system configurations** (3 batch sizes × 2 GPU counts), *not* across repeated runs of the same configuration.

| | MATH500 (n=4) | | | AIME'24 (n=16) | | | AIME'24 (n=64) | | |
|---|---|---|---|---|---|---|---|---|---|
| | **BF16** | **FP16** | **FP32** | **BF16** | **FP16** | **FP32** | **BF16** | **FP16** | **FP32** |
| **DeepSeek-R1-Distill-Qwen-7B** | 0.3158 | 0.1463 | 0.1021 | 1.7151 | 0.8273 | 1.1785 | 0.3749 | 0.5391 | 0.7377 |
| **DeepSeek-R1-Distill-Llama-8B** | 0.3602 | 0.3371 | 0.1211 | 1.5124 | 1.8792 | 0.8606 | 0.8774 | 0.8539 | 0.5034 |
| **Qwen2.5-7B-Instruct** | 0.4663 | 0.1686 | 0.0274 | 0.7056 | 0.2523 | 0 | 0.1784 | 0.1382 | 0 |
| **Llama-3.1-8B-Instruct** | 0.6020 | 0.1725 | 0.3293 | 0.5992 | 0.2282 | 0.7759 | 0.4216 | 0.2898 | 0.1296 |

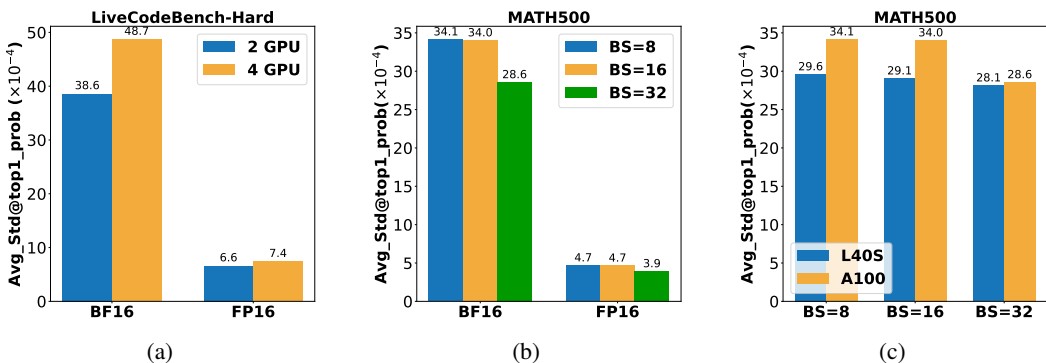

Figure 6: Controlled experiments for Avg_Std@top1_prob under different runtime configurations: (a) **2GPU vs 4GPU** (varying batch size); (b) **BS=8 vs 16 vs 32** (varying A100 GPU Count); (c) **L40S vs A100** (BF16, varying GPU Count). Larger GPU counts and smaller batch sizes tend to increase inference instability.

**of sampling-based evaluations as well.** When using random sampling with temperature $T > 0$, researchers typically report the mean accuracy averaged over multiple runs (commonly referred to as Pass@1). We conduct Pass@1 evaluations across two benchmarks–AIME'24 and MATH500–using 16 and 64 independent sampling runs for AIME'24 and 4 runs for MATH500. Here we emphasize that in Table 5, the reported standard deviation is calculated based on Pass@1 performance across 6 system configurations per model (batch sizes and GPU counts), not across repeated runs of the same configuration. Thus, the reported variance largely reflects the impact from limited numerical precision, not the inherent variance of models. As shown in Table 5, we observe a clear trend: numerical precision introduces an additional source of variance beyond the intended randomness, and lower-precision formats such as BF16 tend to produce higher output variance. See Appendix G for the complete Pass@1 results that support these variance statistics.

This pattern largely holds across the models we evaluate, especially on the MATH500 benchmark. A notable exception arises in the AIME'24 results for DeepSeek-R1-Distill-Qwen-7B, where at n = 64, FP32 exhibits higher variance (0.7377) than BF16 (0.3749). We interpret this as a result of dataset size and sampling dynamics rather than a contradiction of the overall trend. With only 30 problems in AIME'24, a single answer can shift Pass@1 by about 3.33%, amplifying statistical noise. Moreover, BF16–which exhibits the highest variance at n = 16–also shows the greatest improvement when increasing to n = 64. This suggests that instability from reduced numerical precision can be mitigated with sufficient averaging, but remains a dominant factor at typical sample sizes.

These findings highlight numerical precision as a critical factor in the reproducibility of sampling-based evaluations. Researchers using random sampling with BF16 may need substantially more runs to achieve the same statistical confidence as with higher-precision formats, representing a computational overhead that is rarely acknowledged in current evaluation practices.

### 3.4 Ablation: How Runtime Configurations Affect Reproducibility

After establishing that numerical precision plays a crucial role in LLM reproducibility, we now investigate how specific runtime configurations—batch size, number of GPUs, and GPU type—affect

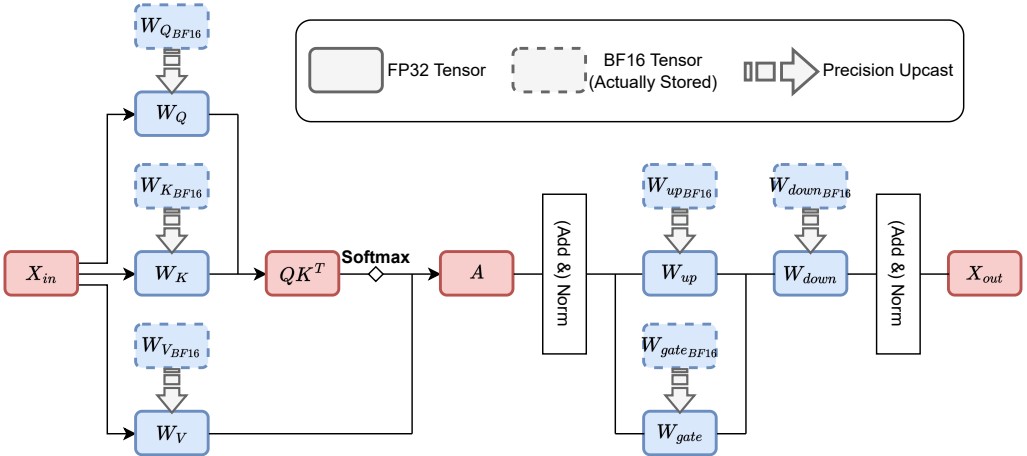

Figure 7: Workflow of LayerCast within one transformer block. Here we omit skip connections, position embeddings and activation functions.

output stability across different precision formats. We conduct experiments of *greedy decoding* setting on DeepSeek-R1-Distill-Qwen-7B, as shown in Figure 6.

Our analysis of runtime configurations reveals three key factors affecting token probability variations. First, in Figure 6 (a), configurations with 4 GPUs tend to exhibit higher probability variation than those with 2 GPUs across 3 tested batch sizes (particularly in BF16 precision), potentially due to increased parallel computation introducing more varied floating-point operation orderings and consequently different rounding errors. Second, Figure 6 (b) suggests that smaller batch sizes counter-intuitively produce higher variance in token probabilities because they may require more sequential processing steps that accumulate rounding errors, while larger batches benefit from parallel computation within optimized CUDA kernels that limit error accumulation. Third, GPU architecture matters: Figure 6 (c) shows A100s generally exhibit slightly higher probability variance than L40S under identical configurations, likely due to differences in hardware-level floating-point implementations and memory hierarchies. All these effects are most pronounced under BF16 precision with its limited mantissa bits making it especially susceptible to rounding effects. For more results of runtime configurations and tested tasks, please refer to the Appendix F.

In summary, our experiments reveal three critical insights about numerical precision and reproducibility in LLM inference. **First**, the fundamental cause of nondeterministic outputs is the small gap between competing logits, which makes token selection vulnerable to minute numerical fluctuations. **Second**, precision format critically impacts stability, with FP32 providing near-perfect determinism, FP16 offering moderate stability, and BF16 exhibiting significant variance despite being commonly used. **Third**, specific runtime configurations—particularly GPU count, batch size, and GPU architecture—further affect reproducibility, with these effects most pronounced in lower precision formats. These findings highlight the urgent need for standardized evaluation practices that account for numerical precision effects, especially as LLMs continue to be deployed in increasingly critical applications where reproducibility is essential.

## 4 Near-Perfect Deterministic Reproduction: LayerCast

Given our findings on when and why reproducibility breaks, we now propose directions to improve reproducibility in LLM inference. The basic solution is using FP32 precision, as we've shown in previous sections. However, this approach incurs significant costs: it doubles the memory usage and inference time compared to BF16, making it impractical for many production environments.

We propose a more efficient solution: LayerCast, a hybrid precision approach that balances computational stability with memory efficiency. LayerCast works by: (1) Loading the model parameters initially in FP32 precision; (2) Explicitly casting all linear layer weights and biases to BF16 for storage before inference; and (3) As inference runs, upcasting each weight back to FP32 just-in-time for matrix multiplication, **one at a time.**

As illustrated in Figure 7, this approach ensures all computations occur in full FP32 precision while storing weights in memory-efficient 16-bit formats. Thus, this model benefits from FP32's stability during computation, while the memory footprint remains closer to that of 16-bit models. This provides determinism comparable to full FP32 inference but with substantially reduced memory requirements, particularly beneficial for the KV cache in long-context scenarios.

Our experimental results strongly support this approach. When examining the standard deviation of accuracy across runs, Layer Cast achieves stability nearly identical to FP32, while BF16 shows much higher variability. As shown in Figure 8, the divergence index measurements further confirm that LayerCast produces consistent outputs across different batch sizes and GPU configurations, with divergence rates below 3.4%. From resource perspective, LayerCast offers substantial benefits over full FP32: memory usage is reduced by 34% (particularly important for KV cache in long-context scenarios). These improvements make LayerCast a practical solution for applications requiring both deterministic outputs and reasonable performance. For full result, please refer to Appendix H.

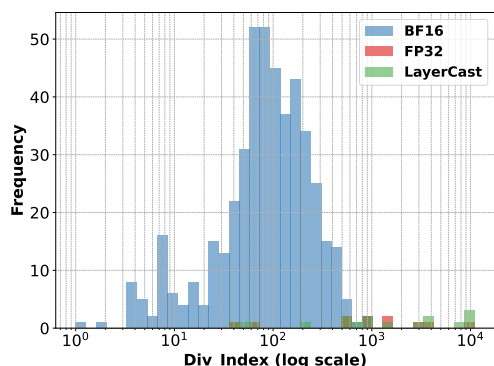

Figure 8: Distribution of Div_Index for DeepSeek-R1-Distill-Qwen-7B on MATH500 under BF16, FP32, and LayerCast.

## 5    Related Works

Since the era of traditional deep learning, the reproducibility of models [28, 39, 33] have remained complex and challenging problems that have yet to be fully resolved. As large language models (LLMs) have risen to prominence, numerous empirical studies [21, 35, 3, 11, 19, 32, 24, 38] have shown that nondeterministic behavior during LLM inference is widely observed. Existing study [5] have found a clear negative correlation between output length and inference consistency: As the length of generated text increases, output variation during inference also rises, which explains the phenomenon that reasoning models tend to exhibit greater inference uncertainty.

Multiple factors can affect the reproducibility of LLM inference results, including but not limited to prompt formatting, decoding parameters (such as temperature and top-p thresholds), random seed settings, and hardware and software configurations [17, 10, 9]. Existing studies [24, 11, 3, 21, 4] have systematically analyzed the impact of decoding parameters (e.g., temperature, top-p) on the stability of LLM inference outputs. Hochlehnert et al. [16] points out that many reported improvements in LLM performance are, in fact, partially attributable to unfair comparisons and unreported sources of variance in the evaluation process. In practical applications, using FP32 (single-precision floating point) inference [30, 31] is often empirically believed to enhance the robustness of numerical computations. Feng et al. [12] investigates the impact of numerical precision on the mathematical reasoning ability of LLMs. However, there is currently a lack of systematic and quantitative studies specifically analyzing the effects of different numerical formats (e.g., FP32, FP16, BF16) on the reproducibility of LLM inference.

## 6    Conclusion

In this paper, we conducted a comprehensive investigation into the reproducibility challenges in LLM inference caused by numerical precision issues. Our experiments across multiple models, tasks, and hardware configurations revealed that even under supposedly deterministic greedy decoding, outputs can vary significantly due to floating-point arithmetic non-associativity. We demonstrated that precision format critically impacts stability, with FP32 providing near-perfect determinism while BF16—despite being widely used—exhibits substantial variance. To address these challenges without incurring the full overhead of FP32, we proposed LayerCast, a practical solution that achieves FP32-level determinism while maintaining reasonable memory efficiency. Our findings highlight the importance of standardizing evaluation practices to account for numerical precision effects, especially as LLMs are increasingly deployed in critical applications where reproducibility is essential.

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

## A  Limitations

Despite the importance of our work in highlighting and addressing numerical precision's effect on reproducibility, there are limitations that are worth noting. First, our experiments focus on a certain set of models and settings. Due to resource constraints, we do not conduct large-scale experiments on much larger models (>70B) or a wide range of GPU architectures or accelerators; these settings might exhibit different levels of numerical behaviors. Additionally, our study primarily addresses numerical precision issues in transformer-based LLMs and may not fully generalize to other architectures or modalities.

## B  Broader Impacts

This research has several societal impacts. First, by raising the discussion of reproducibility issues in LLM inference, our work promotes greater scientific rigor in AI research, calling for a more reliable comparison of models and techniques. This is particularly important as LLMs are increasingly deployed in critical areas like healthcare, education, and public services, where consistency and reproducibility are essential. On the other hand, our findings reveal that true reproducibility may require additional computational resources, potentially exacerbating the already significant environmental effects of LLM development. We hope our work will encouage the AI community to establity better standards of evaluation–ultimately leading to more trustworthy and reliable AI systems.

## C  Prompt Formats Used in Our Evaluation

Our evaluation implementation and prompt formats are adapted from SkyThought repository. AIME'24 and MATH500 share a common prompt format across all models. LiveCodeBench-Easy, LiveCodeBench-Medium, and LiveCodeBench-Hard also have the same prompt format across all models, which varies by the test type of the problems (stdin, functional). The `___PROBLEM_TEXT___` is the placeholder that will be replaced with specific problem texts from benchmarks during prompting.

### C.1  MATH Benchmarks: AIME'24 and MATH500

**DeepSeek-R1-Distill-Qwen-7B and DeepSeek-R1-Distill-Llama-8B**

```
<|begin_of_sentence|><|User|>Return your final response within
↪ \\boxed{}.  ___PROBLEM_TEXT___<|Assistant|><think>\n
```

**Qwen2.5-7B-Instruct**

```
<|im_start|>system\nYou are Qwen, created by Alibaba Cloud.  You are
↪ a helpful assistant.<|im_end|>\n<|im_start|>user\nReturn your
↪ final response within \\boxed{}.  ___PROBLEM_TEXT___<|im_end|>\n
↪ <|im_start|>assistant\n
```

**Llama-3.1-8B-Instruct**

```
<|begin_of_text|><|start_header_id|>system<|end_header_id|>\n\nCutting
↪ Knowledge Date:  December 2023\nToday Date:  26 Jul 2024\n\n
↪ <|eot_id|><|start_header_id|>user<|end_header_id|>\n\nReturn
↪ your final response within \\boxed{}.  ___PROBLEM_TEXT___<|eot_id|>
↪ <|start_header_id|>assistant<|end_header_id|>\n\n
```

## C.2 Code Generation Benchmarks: LiveCodeBench-Easy, LiveCodeBench-Medium, LiveCodeBench-Hard

---

**DeepSeek-R1-Distill-Qwen-7B and DeepSeek-R1-Distill-Llama-8B**

For problems with "stdin" tests:
```
<|begin_of_sentence|><|User|>Generate an executable Python function
↪ generated from the given prompt.  The function should take stdin
↪ as input and print the output.  Simply call the function after
↪ the definition.  ___PROBLEM_TEXT___<|Assistant|><think>\n
```

---

For problems with "functional" tests:
```
<|begin_of_sentence|><|User|>Generate an executable Python function
↪ generated from the given prompt.  Return the function body without
↪ invoking it at the final solution.  ___PROBLEM_TEXT___<|Assistant|>
↪ <think>\n
```

---

**Qwen2.5-7B-Instruct**

For problems with "stdin" tests:
```
<|im_start|>system\nYou are Qwen, created by Alibaba Cloud.  You are
↪ a helpful assistant.<|im_end|>\n<|im_start|>user\nGenerate an
↪ executable Python function generated from the given prompt.  The
↪ function should take stdin as input and print the output.  Simply
↪ call the function after the definition.  ___PROBLEM_TEXT___
↪ <|im_end|>\n<|im_start|>assistant\n
```

---

For problems with "functional" tests:
```
<|im_start|>system\nYou are Qwen, created by Alibaba Cloud.  You are
↪ a helpful assistant.<|im_end|>\n<|im_start|>user\nGenerate an
↪ executable Python function generated from the given prompt.
↪ Return the function body without invoking it at the final
↪ solution.  ___PROBLEM_TEXT___<|im_end|>\n<|im_start|>assistant\n
```

---

**Llama-3.1-8B-Instruct**

For problems with "stdin" tests:
```
<|begin_of_text|><|start_header_id|>system<|end_header_id|>\n\nCutting
↪ Knowledge Date:  December 2023\nToday Date:  26 Jul 2024\n\n
↪ <|eot_id|><|start_header_id|>user<|end_header_id|>\n\nGenerate an
↪ executable Python function generated from the given prompt.  The
↪ function should take stdin as input and print the output.  Simply
↪ call the function after the definition.  ___PROBLEM_TEXT___<|eot_id|>
↪ <|start_header_id|>assistant<|end_header_id|>\n\n
```

---

For problems with "functional" tests:
```
<|begin_of_text|><|start_header_id|>system<|end_header_id|>\n\nCutting
↪ Knowledge Date:  December 2023\nToday Date:  26 Jul 2024\n
↪ <|eot_id|><|start_header_id|>user<|end_header_id|>\n\nGenerate an
↪ executable Python function generated from the given prompt.
↪ Return the function body without invoking it at the final
↪ solution.  ___PROBLEM_TEXT___<|eot_id|><|start_header_id|>assistant
↪ <|end_header_id|>\n\n
```

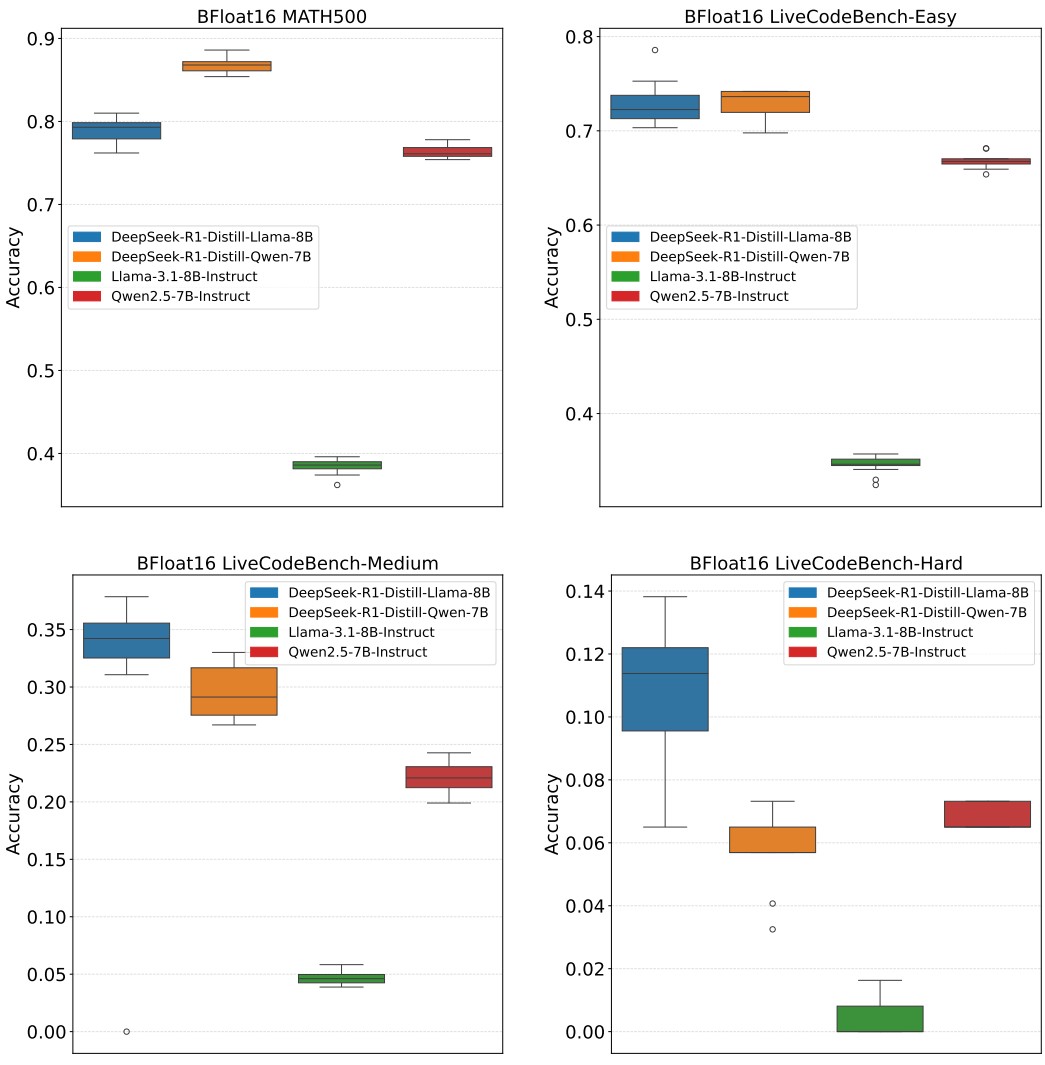

Figure 9: Accuracy varies significantly across different settings under BFloat16.

# D   Supplementary Results on Accuracy Variance under BFloat16

In Section 1, we demonstrated the variance of accuracy on BF16 with the same seed and greedy decoding but different batch sizes and number of GPUs in Figure 1. To provide a comprehensive understanding of reproducibility challenges across different problem domains, Figure 9 extends this analysis to all evaluated datasets. The results reveal a clear pattern: while accuracy variance under BF16 precision is relatively modest for MATH500 and LiveCodeBench-Easy, it becomes substantially more pronounced for the more challenging LiveCodeBench-Medium and LiveCodeBench-Hard benchmarks. This dataset-dependent variation suggests that the impact of numerical precision on reproducibility scales with problem complexity.

# E   Supplementary Results on Greedy Decoding

Following up on Figure 5 and Table 3 from Section 3.2, we present comprehensive results across all experimental settings. We report three key metrics: Std@Acc, Div_Percent, and Average Div_Index.

Table 6 presents the standard deviation of accuracy (Std@Acc) across 12 runtime configurations under BF16, FP16, and FP32 precisions for LiveCodeBench-Medium and LiveCodeBench-Hard datasets. Table 7 reports the average standard deviation of output lengths (Avg_Std@Output_Length)

Table 6: Std@Acc of greedy decoding across 12 different settings (GPU types, GPU counts, and batch sizes) under BF16, FP16, and FP32 numerical precisions on LiveCodeBench datasets

| | LiveCodeBench-Medium | | | LiveCodeBench-Hard | | |
|---|---|---|---|---|---|---|
| | **BF16** | **FP16** | **FP32** | **BF16** | **FP16** | **FP32** |
| **DeepSeek-R1-Distill-Qwen-7B** | 2.28% | 2.21% | 0.44% | 1.19% | 1.98% | 0.42% |
| **DeepSeek-R1-Distill-Llama-8B** | 2.08% | 1.84% | 0.78% | 2.15% | 1.90% | 0.53% |
| **Qwen2.5-7B-Instruct** | 1.44% | 0.24% | 0 | 0.40% | 1.45e-17 | 1.45e-17 |
| **Llama-3.1-8B-Instruct** | 0.70% | 0.48% | 0.44% | 0.65% | 3.62e-18 | 3.62e-18 |

Table 7: Avg_Std@Output_Length of greedy decoding across 12 different settings (GPU types, GPU counts, and batch sizes) under BF16, FP16, and FP32 numerical precisions on LiveCodeBench datasets

| | LiveCodeBench-Medium | | | LiveCodeBench-Hard | | |
|---|---|---|---|---|---|---|
| | **BF16** | **FP16** | **FP32** | **BF16** | **FP16** | **FP32** |
| **DeepSeek-R1-Distill-Qwen-7B** | 8893.02 | 8026.63 | 547.50 | 8036.21 | 7476.98 | 646.30 |
| **DeepSeek-R1-Distill-Llama-8B** | 8240.77 | 7842.29 | 646.30 | 7518.49 | 6827.19 | 690.86 |
| **Qwen2.5-7B-Instruct** | 26.47 | 6.18 | 0 | 36.01 | 9.26 | 0 |
| **Llama-3.1-8B-Instruct** | 60.25 | 7.49 | 0.89 | 78.26 | 15.95 | 2.75 |

across the same 12 runtime configurations and precision settings for these two datasets. In Table 8, we measure the Div_Index of each example across 12 runtime configurations—within the 0–Div_Index range, the 12 responses corresponding to the same problem have exactly the same token at each position—and report the average Div_Index over the entire dataset. Table 9 shows the percentage of examples in each dataset exhibiting divergent outputs across the 12 runtime settings.

These results consistently support our conclusion that during greedy decoding, rounding errors from floating-point operations cause serious reproducibility problems. As numerical precision increases from BF16 to FP32, the divergence phenomenon is substantially reduced, evidenced by lower Std@Acc, lower Div_Percent, and larger Div_Index values.

### E.1 Additional Results on Larger Models and Diverse Tasks

To further validate our findings on the impact of numerical precision on reproducibility, we conducted additional experiments on a larger model (Qwen3-32B) and a diverse reasoning task (GPQA Diamond). These results demonstrate that the reproducibility challenges we identified are not limited to 7-8B models or math/coding tasks.

**Larger Model Evaluation (Qwen3-32B):** To demonstrate the impact of numerical precision on reproducibility for larger models, we benchmarked Qwen3-32B on the AIME'24 dataset under different runtime configurations covering 2 GPU counts (2 and 4), 2 GPU versions (A100 and L40S), and 3 batch sizes (8, 16, 32). Due to memory constraints, Qwen3-32B cannot be loaded into 2 L40S GPUs with FP32 precision, resulting in 9 feasible configurations for FP32 out of the 12 total configurations. The results in Table 10 confirm that larger models exhibit the same reproducibility challenges we identified in 7-8B models. Across all configurations, BF16 shows significant variance with Std@Acc of 5.02%, while FP32 maintains much better consistency with Std@Acc of only 1.11%.

**Diverse Task Evaluation (GPQA Diamond):** To demonstrate that reproducibility issues extend beyond math and coding tasks, we evaluated DeepSeek-R1-Distill-Llama-8B on the GPQA Diamond benchmark, which focuses on graduate-level science questions. Table 11 shows the results across 12 different runtime configurations. The results reveal that BF16 exhibits Std@Acc of 3.03% compared to 1.96% for FP32, confirming that numerical precision affects reproducibility across diverse reasoning domains.

Table 8: Average Div_Index across 12 different settings under BF16, FP16, and FP32 numerical precisions. In the table, a value of "-1" indicates that no divergence occurred for any example in the dataset under the given setting.

| | | DeepSeek-R1-Distill-Qwen-7B | DeepSeek-R1-Distill-Llama-8B | Qwen2.5-7B-Instruct | Llama-3.1-8B-Instruct |
|---|---|---|---|---|---|
| **AIME'24** | **BF16** | 45.67 | 67.37 | 82.37 | 22.44 |
| | **FP16** | 430.13 | 430.47 | 457.41 | 635.43 |
| | **FP32** | -1 | 13520 | -1 | 1976.40 |
| **MATH500** | **BF16** | 65.33 | 76.85 | 103.58 | 38.07 |
| | **FP16** | 395.46 | 463.14 | 267.59 | 284.89 |
| | **FP32** | 1825.79 | 1855.17 | -1 | 1509.61 |
| **LCB-Easy** | **BF16** | 35.11 | 47.43 | 46.98 | 53.99 |
| | **FP16** | 292.99 | 327.53 | 87.53 | 133.80 |
| | **FP32** | 1143.30 | 1036.04 | -1 | 108.86 |
| **LCB-Medium** | **BF16** | 44.23 | 47.59 | 44.23 | 54.33 |
| | **FP16** | 267.81 | 287.83 | 107.07 | 172.73 |
| | **FP32** | 2291.83 | 2585.83 | -1 | 125.00 |
| **LCB-Hard** | **BF16** | 43.85 | 31.57 | 43.64 | 54.95 |
| | **FP16** | 216.16 | 329.38 | 136.89 | 221.76 |
| | **FP32** | 5807.83 | 3832.54 | -1 | 352.78 |

Table 9: Div_Percent across 12 different settings (GPU types, GPU counts, and batch sizes) under BF16, FP16, and FP32 numerical precisions

| | | DeepSeek-R1-Distill-Qwen-7B | DeepSeek-R1-Distill-Llama-8B | Qwen2.5-7B-Instruct | Llama-3.1-8B-Instruct |
|---|---|---|---|---|---|
| **AIME'24** | **BF16** | 100% | 100% | 100% | 53.33% |
| | **FP16** | 100% | 100% | 73.33% | 23.33% |
| | **FP32** | 0 | 6.67% | 6.67% | 16.67% |
| **MATH500** | **BF16** | 99.60% | 100% | 90.20% | 77.60% |
| | **FP16** | 86.00% | 87.80% | 37.60% | 36% |
| | **FP32** | 5.80% | 6.00% | 1.80% | 9.20% |
| **LCB-Easy** | **BF16** | 100% | 100% | 72.53% | 96.15% |
| | **FP16** | 100% | 97.25% | 16.48% | 41.76% |
| | **FP32** | 10.99% | 13.19% | 0 | 3.85% |
| **LCB-Medium** | **BF16** | 100% | 100% | 89.32% | 98.06% |
| | **FP16** | 100% | 100% | 35.44% | 48.06% |
| | **FP32** | 19.90% | 23.30% | 0 | 3.40% |
| **LCB-Hard** | **BF16** | 100% | 100% | 95.12% | 100% |
| | **FP16** | 100% | 100% | 50.41% | 58.54% |
| | **FP32** | 24.39% | 30.08% | 0 | 7.32% |

These additional experiments on a larger model and a diverse reasoning task further validate our core findings: numerical precision critically impacts reproducibility across model scales and task domains, with BF16 consistently showing higher variance compared to FP32.

## E.2 Verification with HuggingFace Transformers Backend

To verify that the reproducibility issues we observed are not specific to the vLLM inference backend, we conducted additional experiments using the standard HuggingFace Transformers implementation without vLLM or PagedAttention. The experimental setup is identical to our main paper: 2 GPU

Table 10: Accuracy of Qwen3-32B on AIME'24 under different runtime configurations (GPU types, GPU counts, and batch sizes) and numerical precisions. OOM indicates out-of-memory errors.

| | 4 GPUs | | | 2 GPUs | | |
|---|---|---|---|---|---|---|
| Precision | A100-BS8 | A100-BS16 | A100-BS32 | A100-BS8 | A100-BS16 | A100-BS32 |
| BF16 | 80.00% | 83.33% | 83.33% | 76.67% | 83.33% | 73.33% |
| FP32 | 76.67% | 76.67% | 76.67% | 76.67% | 80.00% | 76.67% |
| | 4 GPUs | | | 2 GPUs | | |
| Precision | L40S-BS8 | L40S-BS16 | L40S-BS32 | L40S-BS8 | L40S-BS16 | L40S-BS32 |
| BF16 | 73.33% | 83.33% | 70.00% | 83.33% | 76.67% | 73.33% |
| FP32 | 76.67% | 76.67% | 76.67% | OOM | OOM | OOM |

Table 11: Accuracy of DeepSeek-R1-Distill-Llama-8B on GPQA Diamond under different runtime configurations (GPU types, GPU counts, and batch sizes) and numerical precisions.

| | 4 GPUs | | | 2 GPUs | | |
|---|---|---|---|---|---|---|
| Precision | A100-BS8 | A100-BS16 | A100-BS32 | A100-BS8 | A100-BS16 | A100-BS32 |
| BF16 | 33.84% | 38.38% | 41.41% | 40.91% | 41.92% | 41.41% |
| FP32 | 40.91% | 45.45% | 43.43% | 40.91% | 43.43% | 40.91% |
| | 4 GPUs | | | 2 GPUs | | |
| Precision | L40S-BS8 | L40S-BS16 | L40S-BS32 | L40S-BS8 | L40S-BS16 | L40S-BS32 |
| BF16 | 43.43% | 38.38% | 44.95% | 38.89% | 43.94% | 41.92% |
| FP32 | 38.89% | 42.42% | 44.95% | 42.93% | 41.41% | 44.44% |

counts (2 and 4), 2 GPU versions (A100 and L40S), and 3 batch sizes (8, 16, 32), covering 12 settings for each precision.

Table 12 shows the results for two models on AIME'24. The same pattern emerges: FP32 inference is fully reproducible with 0% Std@Acc for both models, while BF16 exhibits Std@Acc of 2.23% for Qwen2.5-7B-Instruct and 1.51% for Llama-3.1-8B-Instruct, similar variance to what we reported with vLLM. These results demonstrate that the reproducibility phenomenon is fundamental to GPU hardware and floating-point arithmetic, and is observable across different inference backends.

Table 12: Std@Acc using HuggingFace Transformers backend (without vLLM) on AIME'24 across 12 different settings, confirming that reproducibility issues are not backend-specific.

| Model / Dataset | Std@Acc BF16 | Std@Acc FP32 |
|---|---|---|
| Qwen2.5-7B-Instruct / AIME'24 | 2.23% | 0 |
| Llama-3.1-8B-Instruct / AIME'24 | 1.51% | 0 |

# F Supplementary Results on Ablation Study

Continuing the discussion from Section 3.4, we provide additional results that isolate specific runtime configurations to examine their individual effects on reproducibility. For greedy decoding experiments, we focus exclusively on BF16 and FP16 precisions, since FP32 precision effectively mitigates numerical-precision-related errors and thus provides limited insight into reproducibility challenges.

Tables 13 and 14 discuss the effect of different numbers of GPUs since, intuitively, the effect has a dependency on the GPU version, so we consider the situation on L40S and A100, separately. Table 13 reports the Avg_Std@top1_prob of LLM inference responses for the same question using 2 or 4 L40S GPUs under three different batch sizes (8, 16, and 32), while Table 14 presents the

Table 13: Instability (Avg_Std@top1_prob ($\times 10^{-4}$)) under different numbers of L40S GPUs

| | | DeepSeek-R1-Distill-Qwen-7B | | DeepSeek-R1-Distill-Llama-8B | | Qwen2.5-7B-Instruct | | Llama-3.1-8B-Instruct | |
|---|---|---|---|---|---|---|---|---|---|
| | | 2GPU | 4GPU | 2GPU | 4GPU | 2GPU | 4GPU | 2GPU | 4GPU |
| **AIME'24** | **BF16** | 29.66 | 24.52 | 23.02 | 28.01 | 29.40 | 34.05 | 22.12 | 44.54 |
| | **FP16** | 5.83 | 5.19 | 3.18 | 3.95 | 2.94 | 3.91 | 2.89 | 7.30 |
| **MATH500** | **BF16** | 32.91 | 32.67 | 23.03 | 27.33 | 26.51 | 24.91 | 35.92 | 42.31 |
| | **FP16** | 4.48 | 4.72 | 3.19 | 3.55 | 3.79 | 3.75 | 4.80 | 5.75 |
| **LCB-Easy** | **BF16** | 42.80 | 46.82 | 28.46 | 34.80 | 28.48 | 30.31 | 28.05 | 33.93 |
| | **FP16** | 6.03 | 6.55 | 4.27 | 4.68 | 4.29 | 4.42 | 3.62 | 4.45 |
| **LCB-Medium** | **BF16** | 38.44 | 44.52 | 30.25 | 39.24 | 29.68 | 34.56 | 27.62 | 36.34 |
| | **FP16** | 7.04 | 7.68 | 4.52 | 5.00 | 4.84 | 4.74 | 3.45 | 4.05 |
| **LCB-Hard** | **BF16** | 38.63 | 48.74 | 29.76 | 36.10 | 34.31 | 29.60 | 28.90 | 34.60 |
| | **FP16** | 6.62 | 7.36 | 5.04 | 5.58 | 5.03 | 4.85 | 3.17 | 5.61 |

Table 14: Instability (Avg_Std@top1_prob ($\times 10^{-4}$)) under different numbers of A100 GPUs

| | | DeepSeek-R1-Distill-Qwen-7B | | DeepSeek-R1-Distill-Llama-8B | | Qwen2.5-7B-Instruct | | Llama-3.1-8B-Instruct | |
|---|---|---|---|---|---|---|---|---|---|
| | | 2GPU | 4GPU | 2GPU | 4GPU | 2GPU | 4GPU | 2GPU | 4GPU |
| **AIME'24** | **BF16** | 32.91 | 33.90 | 31.23 | 25.86 | 30.64 | 41.65 | 53.72 | 35.56 |
| | **FP16** | 7.23 | 5.79 | 4.57 | 3.85 | 4.23 | 3.75 | 5.66 | 5.02 |
| **MATH500** | **BF16** | 34.59 | 30.88 | 31.69 | 25.63 | 26.59 | 21.88 | 43.03 | 44.33 |
| | **FP16** | 5.29 | 4.45 | 4.40 | 3.46 | 4.17 | 3.08 | 7.43 | 6.58 |
| **LCB-Easy** | **BF16** | 48.20 | 43.05 | 36.76 | 32.01 | 28.82 | 34.64 | 36.01 | 35.56 |
| | **FP16** | 7.55 | 6.38 | 6.14 | 4.52 | 4.75 | 4.70 | 5.36 | 4.26 |
| **LCB-Medium** | **BF16** | 44.86 | 44.26 | 42.58 | 37.60 | 31.02 | 35.01 | 39.96 | 33.40 |
| | **FP16** | 8.40 | 7.37 | 6.53 | 5.03 | 5.12 | 4.44 | 4.93 | 4.07 |
| **LCB-Hard** | **BF16** | 46.25 | 47.95 | 41.58 | 37.75 | 36.88 | 39.98 | 41.82 | 33.43 |
| | **FP16** | 7.95 | 6.79 | 6.63 | 5.41 | 5.11 | 4.76 | 5.29 | 5.88 |

corresponding results using A100 GPUs. As shown in Table 13, increasing the number of GPUs from 2 to 4 generally leads to higher Avg_Std@top1_prob. **This observation suggests that increasing the number of GPUs may introduce greater inference nondeterminism.** However, this trend becomes less consistent in the A100 setting as shown in Table 14, where in many cases the 2 GPU results are even slightly higher than those of 4 GPUs. One reason behind it is that A100s do have higher instability in our experiments, which may have more influence beyond the number of GPUs (see Figure 6 (c)).

Table 15 examines the effect of different batch sizes. We report the Avg_Std@top1_prob of LLM inference responses for identical questions under varying GPU counts and GPU types. The results show that **larger batch sizes consistently lead to lower Avg_Std@top1_prob, indicating better reproducibility in model outputs.** This finding aligns with our hypothesis that higher parallelism reduces error accumulation.

Finally, we compare the effect of the two GPU versions we used (L40S and A100). Table 16 reveals that under the same experimental settings, the Avg_Std@top1_prob on the A100 GPU is consistently slightly higher than that on the L40S GPU. **This conforms to the previous findings that, in our experiments, A100 exhibits slightly higher hardware-induced variability, which may contribute to less stable top-1 token predictions across different runtime configurations.**

Tables 17, 18, 19 and 20 present the Std@Acc for each ablation study. Unlike Avg_Std@top1_prob, it's hard to observe clear trends or patterns from Std@Acc. We argue that Avg_Std@top1_prob serves as a more informative metric for ablation studies, as it directly reflects the numerical instability

Table 15: Instability (Avg_Std@top1_prob ($\times 10^{-4}$)) under different batch sizes

| | | DeepSeek-R1-Distill-Qwen-7B | | | DeepSeek-R1-Distill-Llama-8B | | | Qwen2.5-7B-Instruct | | | Llama-3.1-8B-Instruct | | |
| --- | --- | --- | --- | --- | --- | --- | --- | --- | --- | --- | --- | --- | --- |
| | | BS=8 | BS=16 | BS=32 | BS=8 | BS=16 | BS=32 | BS=8 | BS=16 | BS=32 | BS=8 | BS=16 | BS=32 |
| AIME'24 | BF16 | 42.35 | 37.96 | 39.21 | 30.81 | 36.82 | 34.41 | 35.88 | 32.09 | 37.75 | 53.86 | 58.55 | 47.23 |
| | FP16 | 6.66 | 6.17 | 6.16 | 4.40 | 4.14 | 4.10 | 4.03 | 4.49 | 3.91 | 3.79 | 4.96 | 3.41 |
| MATH500 | BF16 | 37.61 | 36.91 | 34.41 | 32.34 | 31.72 | 26.93 | 31.16 | 31.84 | 29.22 | 54.17 | 50.32 | 47.86 |
| | FP16 | 5.26 | 5.23 | 4.79 | 4.28 | 4.19 | 3.68 | 3.87 | 3.96 | 3.57 | 6.69 | 5.59 | 6.14 |
| LCB-Easy | BF16 | 56.00 | 58.99 | 51.39 | 41.93 | 41.56 | 36.80 | 39.14 | 38.27 | 36.21 | 44.63 | 45.88 | 39.60 |
| | FP16 | 7.51 | 7.81 | 7.15 | 5.97 | 5.90 | 5.13 | 5.11 | 5.52 | 4.70 | 5.42 | 4.68 | 4.52 |
| LCB-Medium | BF16 | 53.20 | 51.56 | 48.35 | 45.21 | 46.54 | 41.43 | 41.11 | 39.74 | 39.48 | 47.32 | 46.33 | 43.92 |
| | FP16 | 8.55 | 8.36 | 8.11 | 6.21 | 6.25 | 5.46 | 5.98 | 5.94 | 5.95 | 4.92 | 4.68 | 4.52 |
| LCB-Hard | BF16 | 57.69 | 59.04 | 54.60 | 47.51 | 47.89 | 41.43 | 49.21 | 47.05 | 46.57 | 46.12 | 44.22 | 39.27 |
| | FP16 | 8.41 | 8.50 | 7.74 | 6.59 | 6.54 | 5.90 | 5.96 | 6.01 | 6.15 | 5.54 | 5.75 | 5.30 |

Table 16: Instability (Avg_Std@top1_prob ($\times 10^{-4}$)) under different GPU versions

| | | DeepSeek-R1-Distill-Qwen-7B | | DeepSeek-R1-Distill-Llama-8B | | Qwen2.5-7B-Instruct | | Llama-3.1-8B-Instruct | |
| --- | --- | --- | --- | --- | --- | --- | --- | --- | --- |
| | | A100 | L40S | A100 | L40S | A100 | L40S | A100 | L40S |
| AIME'24 | BF16 | 42.43 | 37.74 | 33.89 | 31.48 | 39.08 | 34.86 | 55.99 | 42.36 |
| | FP16 | 7.22 | 6.34 | 4.72 | 4.23 | 4.69 | 4.34 | 6.86 | 6.01 |
| MATH500 | BF16 | 39.12 | 36.11 | 33.94 | 29.32 | 30.96 | 30.19 | 53.43 | 47.06 |
| | FP16 | 5.50 | 5.09 | 4.51 | 3.87 | 4.22 | 4.00 | 6.86 | 5.58 |
| LCB-Easy | BF16 | 58.36 | 54.88 | 43.33 | 36.87 | 37.45 | 35.12 | 44.84 | 41.29 |
| | FP16 | 7.96 | 7.15 | 6.28 | 5.32 | 5.23 | 4.82 | 5.45 | 4.49 |
| LCB-Medium | BF16 | 54.76 | 48.67 | 47.79 | 43.59 | 38.53 | 37.90 | 46.17 | 42.22 |
| | FP16 | 8.81 | 8.26 | 6.67 | 5.65 | 5.54 | 5.59 | 5.08 | 4.62 |
| LCB-Hard | BF16 | 60.11 | 54.66 | 49.39 | 42.40 | 47.11 | 44.72 | 45.59 | 40.64 |
| | FP16 | 8.38 | 8.10 | 7.04 | 6.18 | 5.91 | 5.91 | 5.59 | 5.13 |

introduced by rounding error—i.e., fluctuations in the predicted probabilities of the same token. However, such fluctuations do not always lead to a token flip, since a flip only occurs when the variation at a specific position exceeds the original top-1 and top-2 probability gap, which is inherently a stochastic event. Moreover, even when a token flip happens—for example, replacing "the" with "that"—it may not necessarily cause the evaluation outcome to change from *correct* to *incorrect* or vice versa, and thus may not affect the final accuracy.

# G   Supplementary Results on Random Sampling

In this section, we show more results related to the discussion in Section 3.3. In the random sampling setting, we report the complete set of Pass@1 results in AIME'24 and MATH500 to further evaluate reproducibility under different precision settings. These experiments are conducted across different batch sizes (8, 16, 32), and GPU counts (2 and 4). The standard deviations are calculated along numeric precision types (BF16, FP16, FP32).

Tables 21, 22, 23, and 24 summarize the Pass@1 accuracies and their standard deviation on DeepSeek-R1-Distill-Qwen-7B, DeepSeek-R1-Distill-Llama-8B, Qwen2.5-7B-Instruct, Llama-3.1-8B-Instruct, respectively. In most cases, FP32 yields the lowest standard deviations, reflecting greater stability across configurations. However, it is worth noticing that there are (4 out of 12) cases where FP16 results are more stable. This suggests that during random sampling, the two sources of randomness can interleave. We still urge researchers to sample more extensively to obtain more reproducible results.

Table 17: Instability (Std@Acc) under different numbers of L40S GPUs

| | | DeepSeek-R1-Distill-Qwen-7B | | DeepSeek-R1-Distill-Llama-8B | | Qwen2.5-7B-Instruct | | Llama-3.1-8B-Instruct | |
|---|---|---|---|---|---|---|---|---|---|
| | | 2GPU | 4GPU | 2GPU | 4GPU | 2GPU | 4GPU | 2GPU | 4GPU |
| **AIME'24** | **BF16** | 3.33% | 13.47% | 5.10% | 1.92% | 1.7e-17 | 1.92% | 0 | 3.33% |
| | **FP16** | 10.72% | 5.10% | 8.39% | 1.92% | 1.7e-17 | 1.7e-17 | 1.92% | 1.92% |
| **MATH500** | **BF16** | 1.10% | 1.11% | 2.53% | 2.16% | 0.90% | 0.72% | 1.70% | 0.23% |
| | **FP16** | 1.79% | 0.64% | 0.90% | 0.46% | 0.40% | 0.46% | 0.46% | 0.23% |
| **LCB-Easy** | **BF16** | 2.54% | 1.14% | 1.93% | 0.64% | 0.64% | 0.95% | 0.83% | 1.27% |
| | **FP16** | 1.14% | 0.32% | 1.10% | 1.45% | 0.32% | 0 | 0.46% | 0.23% |
| **LCB-Medium** | **BF16** | 2.92% | 3.03% | 1.29% | 3.12% | 0.74% | 1.40% | 0.57% | 0.84% |
| | **FP16** | 3.45% | 1.56% | 1.71% | 1.84% | 0.28% | 0.28% | 0.84% | 0.28% |
| **LCB-Hard** | **BF16** | 0.94% | 1.69% | 2.40% | 0.47% | 0 | 0.47% | 0.94% | 0.47% |
| | **FP16** | 2.81% | 0.94% | 2.15% | 3.08% | 0 | 0 | 0 | 0 |

Table 18: Instability (Std@Acc) under different numbers of A100 GPUs

| | | DeepSeek-R1-Distill-Qwen-7B | | DeepSeek-R1-Distill-Llama-8B | | Qwen2.5-7B-Instruct | | Llama-3.1-8B-Instruct | |
|---|---|---|---|---|---|---|---|---|---|
| | | 2GPU | 4GPU | 2GPU | 4GPU | 2GPU | 4GPU | 2GPU | 4GPU |
| **AIME'24** | **BF16** | 5.78% | 5.09% | 3.85% | 8.39% | 1.92% | 1.92% | 1.92% | 1.92% |
| | **FP16** | 5.09% | 1.92% | 5.09% | 6.94% | 1.7e-17 | 1.7e-17 | 0 | 0 |
| **MATH500** | **BF16** | 0.61% | 1.60% | 0.70% | 1.41% | 1.29% | 0.20% | 0.46% | 1.00% |
| | **FP16** | 1.44% | 0.61% | 0.64% | 1.11% | 0.35% | 0.31% | 6.8e-17 | 0.50% |
| **LCB-Easy** | **BF16** | 1.14% | 2.40% | 2.08% | 3.22% | 0.84% | 0.84% | 0.32% | 1.14% |
| | **FP16** | 2.22% | 1.14% | 2.48% | 1.26% | 0.55% | 0.64% | 0.64% | 0.84% |
| **LCB-Medium** | **BF16** | 2.39% | 1.22% | 1.84% | 1.75% | 2.12% | 1.22% | 0.84% | 0.49% |
| | **FP16** | 2.57% | 2.24% | 2.95% | 1.28% | 0 | 0.28% | 0.49% | 0.28% |
| **LCB-Hard** | **BF16** | 0.94% | 0.47% | 1.63% | 0.47% | 0.47% | 0.47% | 0 | 0.47% |
| | **FP16** | 1.88% | 0.47% | 2.05% | 0.82% | 0 | 0 | 0 | 0 |

Table 19: Instability (Std@Acc) under different batch sizes

| | | DeepSeek-R1-Distill-Qwen-7B | | | DeepSeek-R1-Distill-Llama-8B | | | Qwen2.5-7B-Instruct | | | Llama-3.1-8B-Instruct | | |
|---|---|---|---|---|---|---|---|---|---|---|---|---|---|
| | | BS=8 | BS=16 | BS=32 | BS=8 | BS=16 | BS=32 | BS=8 | BS=16 | BS=32 | BS=8 | BS=16 | BS=32 |
| **AIME'24** | **BF16** | 7.20% | 8.61% | 12.58% | 4.19% | 5.00% | 4.19% | 1.92% | 1.67% | 0 | 0 | 2.72% | 1.92% |
| | **FP16** | 7.94% | 5.00% | 4.19% | 7.39% | 6.38% | 3.19% | 0 | 0 | 0 | 1.67% | 1.67% | 0 |
| **MATH500** | **BF16** | 0.76% | 1.30% | 1.12% | 0.50% | 2.07% | 1.91% | 1.00% | 0.69% | 0.85% | 1.44% | 0.44% | 0.93% |
| | **FP16** | 0.77% | 0.82% | 1.30% | 0.35% | 0.97% | 0.50% | 0.43% | 0.35% | 0.33% | 0.35% | 0.35% | 0.16% |
| **LCB-Easy** | **BF16** | 1.80% | 2.12% | 0.78% | 1.82% | 1.45% | 3.11% | 0.45% | 1.30% | 0.32% | 1.22% | 0.52% | 1.29% |
| | **FP16** | 1.38% | 1.34% | 1.00% | 1.45% | 2.70% | 0.95% | 0.69% | 0.28% | 0.53% | 0.94% | 0.64% | 0 |
| **LCB-Medium** | **BF16** | 1.95% | 2.79% | 2.38% | 1.90% | 0.89% | 3.13% | 1.65% | 0.83% | 0.89% | 0.80% | 0.73% | 0.46% |
| | **FP16** | 2.22% | 3.23% | 0.25% | 2.64% | 0.79% | 0.69% | 0.28% | 0.24% | 0.24% | 0.63% | 0.40% | 0.46% |
| **LCB-Hard** | **BF16** | 0.67% | 1.68% | 1.02% | 2.05% | 1.68% | 3.15% | 0.47% | 0.41% | 0.41% | 0.78% | 0 | 0.78% |
| | **FP16** | 2.24% | 2.39% | 1.67% | 2.53% | 2.14% | 1.41% | 0 | 0 | 0 | 0 | 0 | 0 |

# H   Supplementary Results on LayerCast

We evaluate the LayerCast method proposed in Section 4 across multiple models and benchmarks. These evaluations span different batch sizes (8, 16, 32), GPU counts (2 and 4), GPU types (A100 and L40S), and numerical precision settings (BF16, FP32, and LayerCast). All LayerCast experiments follow the same experimental configuration as our main studies.

Table 20: Instability (Std@Acc) under different GPU versions

| | | DeepSeek-R1-Distill-Qwen-7B | | DeepSeek-R1-Distill-Llama-8B | | Qwen2.5-7B-Instruct | | Llama-3.1-8B-Instruct | |
|---|---|---|---|---|---|---|---|---|---|
| | | A100 | L40S | A100 | L40S | A100 | L40S | A100 | L40S |
| AIME'24 | BF16 | 5.44% | 11.81% | 5.87% | 3.44% | 1.82% | 1.72% | 1.72% | 2.11% |
| | FP16 | 3.50% | 7.72% | 5.96% | 6.55% | 1.52e-17 | 1.52e-17 | 0 | 1.72% |
| MATH500 | BF16 | 1.14% | 1.01% | 1.03% | 2.11% | 0.84% | 0.83% | 0.72% | 1.17% |
| | FP16 | 0.99% | 1.21% | 0.83% | 0.64% | 0.33% | 0.39% | 0.35% | 0.34% |
| LCB-Easy | BF16 | 1.69% | 1.79% | 2.66% | 1.42% | 0.90% | 0.75% | 9.59% | 1.08% |
| | FP16 | 1.58% | 0.96% | 1.76% | 2.14% | 0.67% | 0.22% | 0.73% | 0.45% |
| LCB-Medium | BF16 | 1.76% | 2.70% | 2.20% | 2.16% | 1.83% | 1.04% | 0.62% | 0.67% |
| | FP16 | 2.16% | 2.41% | 2.19% | 1.62% | 0.35% | 0.25% | 0.40% | 0.57% |
| LCB-Hard | BF16 | 6.65% | 1.51% | 2.61% | 1.58% | 0.45% | 0.33% | 0.33% | 0.80% |
| | FP16 | 1.82% | 2.14% | 1.52% | 2.38% | 0 | 0 | 0 | 0 |

Table 21: Pass@1 accuracies (%) under 6 system configurations and the standard deviation across them for DeepSeek-R1-Distill-Qwen-7B.

| | 2x A100 | | | 4x A100 | | | Stdev |
|---|---|---|---|---|---|---|---|
| | BS=8 | BS=16 | BS=32 | BS=8 | BS=16 | BS=32 | |
| **AIME'24, n=16** | | | | | | | |
| FP32 | 53.33 | 56.25 | 55.21 | 53.13 | 54.17 | 54.17 | **1.1784** |
| FP16 | 52.71 | 55.00 | 52.92 | 53.33 | 53.13 | 53.13 | **0.8273** |
| BF16 | 53.33 | 56.88 | 55.42 | 56.67 | 53.54 | 53.13 | **1.7151** |
| **AIME'24, n=64** | | | | | | | |
| FP32 | 54.38 | 54.12 | 53.39 | 54.74 | 54.38 | 55.63 | **0.7377** |
| FP16 | 54.27 | 54.06 | 55.52 | 54.38 | 54.11 | 54.38 | **0.5391** |
| BF16 | 54.74 | 54.17 | 55.10 | 54.90 | 55.16 | 55.10 | **0.3749** |
| **MATH500, n=4** | | | | | | | |
| FP32 | 90.60 | 90.80 | 90.80 | 90.75 | 90.90 | 90.70 | **0.1021** |
| FP16 | 90.25 | 90.20 | 90.20 | 90.45 | 90.50 | 90.15 | **0.1463** |
| BF16 | 90.65 | 91.15 | 90.40 | 90.95 | 90.85 | 90.35 | **0.3158** |

It is important to note that LayerCast's effectiveness is not merely empirical but has strong theoretical foundations. As we explain in Section 4, LayerCast performs all computations in FP32 precision, which fundamentally addresses the root cause of non-determinism we identified—the accumulation of rounding errors from lower-precision arithmetic. The only difference from pure FP32 inference is the just-in-time casting from BF16 storage to FP32 for computation, which is a deterministic operation that introduces no additional variance. The experiments we present here serve to demonstrate the practical applicability of this approach across different model architectures.

Table 25 reports the accuracy and standard deviation for DeepSeek-R1-Distill-Qwen-7B across five benchmarks. To facilitate comparison, results under FP32 and BF16 are also included. As expected, BF16 exhibits the most instability, with significantly higher standard deviations. LayerCast matches FP32 in terms of stability (often with zero or near-zero std) while preserving memory efficiency.

To further demonstrate LayerCast's general effectiveness across different model architectures, Table 26 presents results for DeepSeek-R1-Distill-Llama-8B on the AIME'24 benchmark across all 12 runtime configurations. These results confirm that LayerCast consistently achieves perfect reproducibility with 0% Std@Acc (matching FP32) while maintaining memory efficiency benefits, following the same pattern observed with DeepSeek-R1-Distill-Qwen-7B. In contrast, BF16 shows significant variance with Std@Acc of 5.87% on A100 configurations and 3.44% on L40S configurations.

Table 22: Pass@1 accuracies (%) under 6 system configurations and the standard deviation across them for DeepSeek-R1-Distill-Llama-8B.

| | 2x A100 | | | 4x A100 | | | Stdev |
|---|---|---|---|---|---|---|---|
| | BS=8 | BS=16 | BS=32 | BS=8 | BS=16 | BS=32 | |
| **AIME'24, n=16** | | | | | | | |
| FP32 | 43.33 | 43.75 | 41.88 | 42.92 | 43.96 | 42.08 | **0.8606** |
| FP16 | 43.96 | 45.63 | 43.75 | 41.04 | 40.83 | 42.08 | **1.8792** |
| BF16 | 46.46 | 45.42 | 42.71 | 43.13 | 43.54 | 43.13 | **1.5124** |
| **AIME'24, n=64** | | | | | | | |
| FP32 | 43.65 | 43.39 | 44.22 | 44.48 | 43.33 | 44.32 | **0.5034** |
| FP16 | 43.49 | 44.22 | 44.17 | 42.87 | 42.60 | 42.14 | **0.8539** |
| BF16 | 43.13 | 42.76 | 43.07 | 41.41 | 41.98 | 43.85 | **0.8774** |
| **MATH500, n=4** | | | | | | | |
| FP32 | 83.85 | 83.95 | 83.75 | 83.65 | 83.95 | 83.75 | **0.1211** |
| FP16 | 84.55 | 83.95 | 83.95 | 84.15 | 84.25 | 83.55 | **0.3371** |
| BF16 | 84.50 | 84.80 | 84.35 | 84.60 | 85.20 | 85.20 | **0.3602** |

Table 23: Pass@1 accuracies (%) under 6 system configurations and the standard deviation across them for Qwen2.5-7B-Instruct.

| | 2x A100 | | | 4x A100 | | | Stdev |
|---|---|---|---|---|---|---|---|
| | BS=8 | BS=16 | BS=32 | BS=8 | BS=16 | BS=32 | |
| **AIME'24, n=16** | | | | | | | |
| FP32 | 11.46 | 11.46 | 11.46 | 11.46 | 11.46 | 11.46 | **0** |
| FP16 | 11.04 | 11.67 | 11.46 | 11.25 | 11.67 | 11.25 | **0.2523** |
| BF16 | 11.25 | 10.21 | 11.04 | 9.38 | 11.04 | 10.42 | **0.7056** |
| **AIME'24, n=64** | | | | | | | |
| FP32 | 11.25 | 11.25 | 11.25 | 11.25 | 11.25 | 11.25 | **0** |
| FP16 | 10.99 | 11.25 | 11.25 | 11.20 | 11.41 | 11.30 | **0.1382** |
| BF16 | 11.20 | 11.04 | 11.41 | 11.04 | 11.30 | 10.94 | **0.1784** |
| **MATH500, n=4** | | | | | | | |
| FP32 | 74.85 | 74.85 | 74.85 | 74.80 | 74.80 | 74.80 | **0.0274** |
| FP16 | 74.50 | 74.90 | 74.80 | 75.00 | 74.80 | 74.75 | **0.1686** |
| BF16 | 90.65 | 91.15 | 90.40 | 90.95 | 90.85 | 90.35 | **0.3158** |

These results corroborate our main findings in Section 4, demonstrating that LayerCast delivers reproducibility on par with FP32 across different model architectures while operating within a lower memory footprint.

Table 24: Pass@1 accuracies (%) under 6 system configurations and the standard deviation across them for Llama-3.1-8B-Instruct.

| | 2x A100 | | | 4x A100 | | | Stdev |
|---|---|---|---|---|---|---|---|
| | BS=8 | BS=16 | BS=32 | BS=8 | BS=16 | BS=32 | |
| **AIME'24, n=16** | | | | | | | |
| FP32 | 5.00 | 5.42 | 3.75 | 5.00 | 5.42 | 3.75 | **0.7759** |
| FP16 | 5.00 | 5.00 | 4.58 | 5.21 | 5.00 | 5.21 | **0.2282** |
| BF16 | 3.96 | 4.38 | 4.79 | 5.42 | 5.21 | 5.42 | **0.5992** |
| **AIME'24, n=64** | | | | | | | |
| FP32 | 4.01 | 4.01 | 4.27 | 4.01 | 4.06 | 4.27 | **0.1296** |
| FP16 | 3.91 | 4.06 | 4.17 | 4.22 | 4.58 | 4.64 | **0.2898** |
| BF16 | 4.01 | 3.70 | 4.32 | 4.01 | 4.90 | 3.91 | **0.4216** |
| **MATH500, n=4** | | | | | | | |
| FP32 | 37.00 | 36.50 | 36.30 | 36.80 | 36.30 | 36.15 | **0.3293** |
| FP16 | 37.10 | 36.60 | 37.00 | 36.90 | 36.95 | 37.00 | **0.1725** |
| BF16 | 36.25 | 36.95 | 36.65 | 35.35 | 35.90 | 36.75 | **0.6020** |

Table 25: Accuracy and standard deviation of DeepSeek-R1-Distill-Qwen-7B under different GPU counts, precisions (including LayerCast), and batch sizes across 5 benchmarks.

| Benchmark | 2x A100 | | | 4x A100 | | | Std@Acc |
|---|---|---|---|---|---|---|---|
| | BS=8 | BS=16 | BS=32 | BS=8 | BS=16 | BS=32 | |
| **AIME'24** | | | | | | | |
| LayerCast | 0.4333 | 0.4333 | 0.4333 | 0.4333 | 0.4333 | 0.4333 | **0** |
| FP32 | 0.4333 | 0.4333 | 0.4333 | 0.4333 | 0.4333 | 0.4333 | **0** |
| BF16 | 0.4667 | 0.4667 | 0.3667 | 0.4333 | 0.5333 | 0.4667 | **0.0544** |
| **MATH500** | | | | | | | |
| LayerCast | 0.8680 | 0.8680 | 0.8680 | 0.8660 | 0.8660 | 0.8660 | **0.0011** |
| FP32 | 0.8680 | 0.8680 | 0.8680 | 0.8680 | 0.8660 | 0.8700 | **0.0013** |
| BF16 | 0.8700 | 0.8620 | 0.8580 | 0.8540 | 0.8860 | 0.8700 | **0.0114** |
| **LCB-Easy** | | | | | | | |
| LayerCast | 0.7308 | 0.7308 | 0.7363 | 0.7363 | 0.7418 | 0.7418 | **0.0049** |
| FP32 | 0.7363 | 0.7363 | 0.7363 | 0.7308 | 0.7363 | 0.7363 | **0.0022** |
| BF16 | 0.7198 | 0.7418 | 0.7253 | 0.7418 | 0.6978 | 0.7363 | **0.0169** |
| **LCB-Medium** | | | | | | | |
| LayerCast | 0.3058 | 0.3010 | 0.2961 | 0.2961 | 0.3010 | 0.3058 | **0.0043** |
| FP32 | 0.3058 | 0.3010 | 0.3058 | 0.3058 | 0.3058 | 0.3010 | **0.0025** |
| BF16 | 0.2816 | 0.2767 | 0.3204 | 0.2961 | 0.2864 | 0.2718 | **0.0176** |
| **LCB-Hard** | | | | | | | |
| LayerCast | 0.0488 | 0.0488 | 0.0569 | 0.0569 | 0.0488 | 0.0569 | **0.0044** |
| FP32 | 0.0488 | 0.0488 | 0.0488 | 0.0569 | 0.0569 | 0.0569 | **0.0044** |
| BF16 | 0.0569 | 0.0732 | 0.0569 | 0.0650 | 0.0569 | 0.0650 | **0.0066** |

Table 26: Accuracy of DeepSeek-R1-Distill-Llama-8B on AIME'24 under different GPU counts, GPU types, precisions (including LayerCast), and batch sizes.

| Precision | 2x A100 | | | 4x A100 | | |
|---|---|---|---|---|---|---|
| | BS=8 | BS=16 | BS=32 | BS=8 | BS=16 | BS=32 |
| LayerCast | 36.67% | 36.67% | 36.67% | 36.67% | 36.67% | 36.67% |
| FP32 | 36.67% | 36.67% | 36.67% | 36.67% | 36.67% | 36.67% |
| BF16 | 30.00% | 30.00% | 36.67% | 23.33% | 40.00% | 30.00% |
| | 2x L40S | | | 4x L40S | | |
| | BS=8 | BS=16 | BS=32 | BS=8 | BS=16 | BS=32 |
| LayerCast | 36.67% | 36.67% | 36.67% | 36.67% | 36.67% | 36.67% |
| FP32 | 36.67% | 36.67% | 36.67% | 36.67% | 36.67% | 36.67% |
| BF16 | 30.00% | 36.67% | 26.67% | 33.33% | 30.00% | 30.00% |

