# OpenReview forum: "Understanding and Mitigating Numerical Sources of Nondeterminism in LLM Inference"
_NeurIPS.cc/2025/Conference — NeurIPS 2025 oral_

### Official Review · Reviewer_3spC · 2025-06-13

**Clarity:** 3
**Significance:** 4
**Originality:** 3
**Rating:** 5
**Confidence:** 5

**Summary:**

The manuscript highlights the significance of numerical precision for precise LLM reasoning reproducibility. Hardware and system-level heterogeneity result in non-associative property floating-point arithmetic. While most of the researches overestimate this model's uncertainty, making it difficult to distinguish real improvements or random variation. BF16 precision behaves higher risk at numerical sensitivity. Authors conduct experiments on 12 different configurators to highlight that up to 9% of variance would occur even with identical experiment setups.

**Questions:**

1. The relevant results in appendix shows "A100 exhibits slightly higher hardware-induced variability". What is the biggest factor that makes two GPU types exhibit different numerical non-deterministic?
2. The observation "for reasoning models, the token probability differences between the top two competing tokens are often minimal." seems kind of in contrary to the hypothesis in abs "This issue is especially pronounced in reasoning models, where minor rounding differences in early tokens can cascade into divergent chains of thought, ultimately affecting accuracy." Do you think reasoning models would suffer more from the numerical non-deterministic? Table 5 seems to have the conclusion that reasoning model has better stability under BF16.
3. While the LayerCast approach provides a practical compromise between memory efficiency and reproducibility by performing computations in FP32 while storing weights in BF16, similar techniques have been adopted in prior systems such as NVIDIA Transformer Engine and Megatron-LM (e.g., float32_matmul). The paper could benefit from better positioning LayerCast within this line of existing work, and clarifying whether its novelty lies in system implementation, evaluation scale, or reproducibility focus.

**Ethical Concerns:**

["NO or VERY MINOR ethics concerns only"]

**Final Justification:**

Authors rebuttal have adequately resolve my concerns. I believe this paper will be helpful to the NeurIPS and MLSys communities.

**Limitations:**

Authors provide the limitation statement in the start of Appendix. But I still suggest put it on the tail of the paper to broader its visibility if it does not exceed the page limit.

**Paper Formatting Concerns:**

There seems an abnormal '\n' in OpenReview abstract.

**Quality:**

3

**Strengths And Weaknesses:**

**Strengths**
+ Extensive analysis of how numerical precision affects reproducibility.
+ Comprehensive background&motivation illustration, and clear comparison between different numerical precision. The narrative is attractive, motivation is solid and convincing.

**Weaknesses**
- I personally feel reasonable the claim "different hardware implementations or computation orders can lead to slightly different results". However, please add more context to make it more convincing, e.g., more in-depth comparisons or at least references in section 2.2.
- The experimental results in terms of reasoning model seem counterintuitive.
- The proposed LayerCase, compute in FP32 while store in BP16, seems a common practice in LLM serving frameworks. Thereby, the design contribution of this manuscript is quite minor.

---

> ### Author Rebuttal · Authors · 2025-07-31
>
> ### [W1] `Need more context on the root of phenomenon` — We agree and will add more details and references to strengthen this section.
>
> Thank you for this suggestion. We agree that strengthening this context is important. In the current version of Section 2.2, we introduce the concept of non-associativity in floating-point arithmetic and provide a concrete numerical example in Table 2 to illustrate how summation order can change results in both FP32 and BF16.
>
> To make this more convincing as requested, we will expand this section in the final paper. We will add a more detailed discussion on how parallel reduction algorithms on GPUs can lead to different operation orderings, and we will include references to official documentation, such as the PyTorch guide on numerical accuracy [1], which explicitly mentions: *"bitwise identical results are not guaranteed across PyTorch releases, individual commits, or different platforms"*. This will better ground our claims in established computer architecture and deep learning framework principles.
>
> [1] PyTorch Docs: Numerical Accuracy
>
>
> ### [W2 & Q2] `Are reasoning models more or less stable?` — Reasoning models ARE more vulnerable; apparent contradictions arise from different evaluation contexts.
>
> Thanks for bringing up this question. We first clarify that reasoning models are more susceptible to numerical non-determinism precisely *because* they exhibit minimal probability gaps between competing tokens (as shown in Figure 3). When top candidate tokens have nearly identical probabilities, tiny numerical fluctuations from floating-point errors can change their ranking, causing "token flips." These flips, especially early in long chains of thought, cascade into completely different reasoning paths.
>
> Then, regarding the experimental results. While Table 5 in our Appendix reports larger variance of BF16 on reasoning models, do you reference Table 4 in our submission? In Table 4, Qwen2.5-7B-Instruct seems to have higher performance variance than DeepSeek-R1-Distill-Qwen-7B. Here, we provide more results. The tables below show standard deviation of Pass@1 performance (%).
>
> | MATH500 (n=4) | BF16 | FP16 | FP32 |
> | ------- | -------- | -------- | -------- |
> | DeepSeek-R1-Distill-Qwen-7B     | 0.3158    | 0.1463     | 0.1021     |
> | DeepSeek-R1-Distill-Llama-8B     | 0.3602    | 0.3371     | 0.1211     |
> | Qwen2.5-7B-Instruct     | 0.4663    | 0.1686     | 0.0274     |
> | Llama-3.1-8B-Instruct     | 0.6020    | 0.1725     | 0.3293     |
>
> | AIME24 (n=16) | BF16 | FP16 | FP32 |
> | ------- | -------- | -------- | -------- |
> | DeepSeek-R1-Distill-Qwen-7B     | 1.7151    | 0.8273     | 1.1785     |
> | DeepSeek-R1-Distill-Llama-8B     | 1.5124    | 1.8792     | 0.8606     |
> | Qwen2.5-7B-Instruct     | 0.7056    | 0.2523     | 0     |
> | Llama-3.1-8B-Instruct     | 0.5992    | 0.2282     | 0.7759     |
>
> | AIME24 (n=64) | BF16 | FP16 | FP32 |
> | ------- | -------- | -------- | -------- |
> | DeepSeek-R1-Distill-Qwen-7B     | 0.3749    | 0.5391     | 0.7377     |
> | DeepSeek-R1-Distill-Llama-8B     | 0.8774    | 0.8539     | 0.5034     |
> | Qwen2.5-7B-Instruct     | 0.1784    | 0.1382     | 0     |
> | Llama-3.1-8B-Instruct     | 0.4216    | 0.2898     | 0.1296     |
>
> Based on all results, we clarify (i) why the MATH500 slice shows smaller variability for some reasoning model configurations, and (ii) why we still think that reasoning models are more vulnerable to numeric error.
>
> First, the MATH500 evaluation used four trials for pass@1 performance, whereas AIME24 had 16. The relatively fewer trials makes the standard deviation more sensitive to the randomness of random sampling generation: a single flip can move the accuracy by 0.2% while these standard deviation are mostly below 0.5%. The isolated low-variance numbers for MATH500 are from the sampling randomness when there are limited trials, not from an inherent robustness.
>
> Second, wo found evidence in follow-up experiments on AIME24 with n=64. When we increase n of Pass@1 metric, we witness a significant overall drop of the standard deviation, especially in BF16 settings. This demonstrates how repeated independent trials smooth out the intrinsic randomness during the sampling process. As shown in our results, when n=64, i.d., when more sampling randomness is skimmed and numerical error starts dominating the standard deviation, reasoning models consistently show higher variance.
>
> To sum up, we think reasoning models display greater numerical non-determinism, and we recognize that, with limited experiments, there could be isolated results showing some discrepancy.
>
>
> ### [W3 & Q3] `LayerCast is not novel` — Our novelty lies in framing reproducibility as a scientific crisis and providing an accessible, targeted solution for researchers.
>
> We appreciate the reviewer placing our work in the context of prior systems. We agree that the concept of mixed-precision computation is not new. However, the novelty of our work and LayerCast lies in the **problem framing, evaluation focus, and practical accessibility for the research community.**
>
> Our main contribution is not the invention of mixed-precision techniques, but rather the **first systematic quantification of the reproducibility crisis in LLM reasoning evaluation** and the presentation of LayerCast as a direct, targeted solution. While frameworks like Transformer Engine use similar concepts for optimizing training/serving throughput, our work focuses on reproducibility for scientific evaluation. We frame numerical non-determinism not as a performance issue, but as a fundamental threat to reliable benchmarking.
>
> Our novelty can be summarized as:
> 1.  **Reproducibility Focus:** We are the first to evaluate and propose this technique explicitly as a solution to the *reproducibility problem* in LLM evaluation, supported by extensive experiments quantifying the instability of standard methods (Tables 1, 2, 4).
> 2.  **Evaluation Scale:** Our contribution lies in the comprehensive analysis that demonstrates the severity of this problem across different hardware, models, and tasks, proving why such a solution is necessary for reliable research.
> 3.  **Accessibility:** We provide LayerCast as a simple, open-source patch to a widely-used inference library (vLLM). This makes it easy for any researcher to adopt and ensure their results are reproducible, democratizing a practice that was previously embedded in large, complex training frameworks.
>
> We will revise the wording to better highlight our contributions.
>
>
> ### [Q1] `What causes different variability across GPU types?` — We hypothesize it's a combination of kernel implementation, thread scheduling, and compiler optimizations.
>
> This is a very insightful question. While a definitive answer would require proprietary details from the hardware manufacturer, we can offer a hypothesis based on known architectural principles. The difference in variability between an A100 (Ampere architecture) and an L40S (Ada Lovelace architecture) likely stems from a combination of factors at the hardware and software level that influence the order of floating-point operations.
>
> The most probable factors are:
> 1. **Kernel Implementation is different:** Modern low-level kernel libary will use different kernel implementation specialized for different hardware architecture. For example, FlashAttention-3 will use warpgroup-wide WGMMA instruction of Hopper H100 GPUs, which is not available on Ampere architecture like A100. Thus they inherently have different numerical stability [FA3].
> 2.  **Thread Scheduling and Parallelism:** The two GPU architectures have different numbers of SMs, cache hierarchies, and memory bandwidths. The CUDA scheduler may therefore produce different parallel execution plans (i.e., different orderings for associative operations) to best utilize the resources of each specific architecture.
> 3.  **Compiler and Driver Optimizations:** The NVIDIA driver and CUDA compiler generate architecture-specific microcode. The low-level instructions compiled for an A100 can be different from those for an L40S, even from the same source code, leading to different computation paths.

---

> > ### Comment · Reviewer_3spC · 2025-08-01
> > **Thank you for the response**
> >
> > Thank you for the detailed response, especially the clarification on reasoning models and the discussion on variability. They have adequately resolve my concerns. I believe this paper will be helpful to the NeurIPS and MLSys communities, and I have updated my score to support your work.

---

### Official Review · Reviewer_mWEn · 2025-06-22

**Clarity:** 3
**Significance:** 4
**Originality:** 4
**Rating:** 5
**Confidence:** 4

**Summary:**

This paper uncovers a possible reason for pronounced challenges in reproducibility of LLM performance. The authors posit that imprecisions arising from floating point representations being neglected in evaluation benchmarks and practices is a primary cause. They verify this hypothesis through systematic controlled experiments, and identify conditions when model outputs diverge. The authors also develops an inference pipeline termed LayerCast, which is shown to balance memory efficiency with numerical stability.

**Questions:**

Q1: Experiments demonstrating the effectiveness of LayerCast can be more thorough and examine other models. Even in the Appendix, results are reported only for DeepSeek-R1-Distil-Qwen-7B across the five benchmarks. Could the authors comment on this?

**Ethical Concerns:**

["NO or VERY MINOR ethics concerns only"]

**Final Justification:**

I believe that this is a very well-written and presented paper addressing an interesting and important problem. Examining the authors' response reaffirms my initial views on this work.

**Limitations:**

Addressed in the Appendix.

**Paper Formatting Concerns:**

None.

**Quality:**

4

**Strengths And Weaknesses:**

Strengths:

(+) Revealing numerical (im)precision as a cause for lack of reproducibility is deceptively simple- in hindsight it should seem obvious. A strength of this paper is the lengths to which the authors have gone to carefully and systematically demonstrate that this is indeed the case.

(+) The illustrative examples presented demonstrating violation of the associative property for FP addition does an excellent job of highlighting the significance of the problem when scaled up to larger models including LLMs.

(+) Experimental evaluations are presented on reasoning and non-reasoning models, across multiple benchmarks and runtime configurations, demonstrating thoroughness.

(+) This work has the potential for long-lasting impact in any field in which reproducibility heavily relies on precision and representations used in compute components.

(+) Beyond identifying the problem, the paper proposes a possible fix where all computations are performed in FP32, while model parameters continue to be stored in BF16.

Areas for Improvement:

(-) Experiments demonstrating the effectiveness of LayerCast can be more thorough and examine other models. Even in the Appendix, results are reported only for DeepSeek-R1-Distil-Qwen-7B across the five benchmarks.

---

> ### Author Rebuttal · Authors · 2025-07-31
>
> ### [W1 & Q1] `LayerCast experiments need to be more thorough` — We agree and are running new experiments on more models.
>
> We thank the reviewer for this valuable feedback. We acknowledge that in the initial submission, LayerCast results were primarily demonstrated on DeepSeek-R1-Distill-Qwen-7B across five benchmarks. We agree that evaluating LayerCast on additional models would strengthen our claims about its general effectiveness. In response to this suggestion, we are conducting additional experiments with LayerCast on DeepSeek-R1-Distill-Llama-8B across the same benchmarks. Our preliminary results on AIME24 benchmark show that **LayerCast consistently achieves near-FP32 levels of reproducibility** while maintaining the memory efficiency benefits we described. Full results are to be included in the final version.
>
> | Precision | 2A100-BS8 | 2A100-BS16 | 2A100-BS32 | 4A100-BS8 | 4A100-BS16 | 4A100-BS32 | Std@Acc |
> | ------- | -------- | -------- | -------- | -------- | -------- | -------- | -------- |
> | LayerCast     | 0.3667    | 0.3667     | 0.3667     | 0.3667     | 0.3667     | 0.3667     | 0 |
> | FP32     | 0.3667    | 0.3667     | 0.3667     | 0.3667     | 0.3667     | 0.3667     | 0 |
> | BF16     | 0.3    | 0.3     | 0.3667     | 0.2333     | 0.4     | 0.3     | 5.87% |
>
> | Precision | 2L40S-BS8 | 2L40S-BS16 | 2L40S-BS32 | 4L40S-BS8 | 4L40S-BS16 | 4L40S-BS32 | Std@Acc |
> | ------- | -------- | -------- | -------- | -------- | -------- | -------- | -------- |
> | LayerCast     | 0.3667    | 0.3667     | 0.3667     | 0.3667     | 0.3667     | 0.3667     | 0 |
> | FP32     | 0.3667    | 0.3667     | 0.3667     | 0.3667     | 0.3667     | 0.3667     | 0 |
> | BF16     | 0.3    | 0.3667     | 0.2667     | 0.3333     | 0.3     | 0.3     | 3.44% |
>
> It's important to note that LayerCast's effectiveness is not merely empirical but has strong theoretical foundations. As we explain in Section 4, LayerCast performs all computations in FP32 precision, which fundamentally addresses the root cause of non-determinism we identified—the accumulation of rounding errors from lower-precision arithmetic. The only difference from pure FP32 inference is the just-in-time casting from BF16 storage to FP32 for computation, which is a deterministic operation that introduces no additional variance. The additional experiments we're conducting serve to demonstrate the practical applicability of this approach across different model architectures.

---

> > ### Comment · Reviewer_mWEn · 2025-07-31
> > **Thank You Authors**
> >
> > I thank the authors for their careful response to my concerns. I have examined their response to my and other reviewers' comments. I will retain my score, and believe that this paper will be of great interest to the audience at NeurIPS should it be accepted.

---

### Official Review · Reviewer_EEhu · 2025-06-30

**Clarity:** 4
**Significance:** 3
**Originality:** 1
**Rating:** 4
**Confidence:** 4

**Summary:**

The paper analyzes the effect of the known precision problem for floating point numbers as used in numerical computation and the introduced randomness via non-associativity of operations on LLMs and reasoning models' performance metrics. Different inference settings with different GPUs, GPU count, and batch sizes reveal that the common BF16 data type is prone to large deviations in exhibited performance. The authors propose the use of their LayerCast wrapper that keeps memory footprint low by using BF16 weights (for inference), while using the more stable FP32 for computation.

**Questions:**

- Isn't FP32 up-conversion during compute already performed for matrix multiplications on certain hardware (BF16 tensor core acceleration)?

- Does your analysis imply that also during training FP32 should be superior?

- How is precision typically handled for the three common aggregation operations in Transformers: Matrix Multiplication, Layer/RMS Normalization and Self Attention (FlashAttention)?

- Could you provide an inverse benchmark setting, where models are exactly the same and should run reproducibly, but people can report results using the local hardware / library setup - revealing differences? This would necessitate adding some code along with the publication.

**Ethical Concerns:**

["NO or VERY MINOR ethics concerns only"]

**Final Justification:**

While this analysis does not provide fundamentally new insights, it solidly addresses one main concern of scientific work: reproducibility. And it actually shows that for different hardware setups and precision settings, outputs vary by significant margin. While there is still some room for improvement regarding the exact causes of the problem, in principle I rather recommend this work for acceptance.

**Limitations:**

The limitations of the work are clearly stated.

**Paper Formatting Concerns:**

None.

**Quality:**

3

**Strengths And Weaknesses:**

Strengths:
- provides an empirical study of the actual effects of precision problems, and their non-negligible strength compared to differences between model performances
- LayerCast mitigates the precision problems without additional memory overhead

Weaknesses:
- FP32 inference can be far slower on modern hardware
- BF16 tensor core ops actually use higher precision internally and store results in FP32, so this should partly be mitigated already?
- There is no clear analysis how this actually does not badly affect training (which happens typically the other way round: FP32 weights, BF16 computation)
- there is no discussion of actual hardware implementations / algorithms (FlashAttention) wrt the choice of precisions, despite the focus on Attention-based models

---

> ### Author Rebuttal · Authors · 2025-07-31
>
> ### [W1] `FP32 is too slow` — We agree that this is a trade-off, and we proposed LayerCast to help.
>
> We agree with the reviewer that FP32 inference carries a higher computational cost, and this is a **trade-off between speed and precision**. Our goal is not to advocate for abandoning 16-bit formats entirely, but to demonstrate that for applications where reproducibility is critical, the cost of numerical non-determinism in BF16 can be unacceptably high, leading to misleading or irreproducible results (Tables 1 & 2).
>
> This is precisely the motivation behind our proposed solution, LayerCast (Section 4). LayerCast is designed to provide the near-perfect reproducibility of FP32 while mitigating the performance and memory overhead. It achieves this by storing weights in BF16 but performing all computations in FP32, offering a practical compromise for achieving reproducible reasoning without the full cost of FP32 inference.
>
> ### [W2 & Q1] `Isn't BF16 compute already mitigated by internal FP32 accumulation?` — This helps but is insufficient for end-to-end reproducibility due to intermediate casting.
>
> The reviewer rightly notes that modern GPU tensor cores use FP32 for intermediate accumulation during BF16 matrix multiplications, then downcast the results to FP16/BF16 to limit error accumulation. This internal up-conversion does help reduce rounding errors within a single matrix multiplication.
>
> However, as our paper demonstrates, **this is not sufficient to ensure end-to-end reproducibility**. A Transformer forward pass consists of many sequential operations. While a single operation (such as GEMM) may use a higher-precision accumulator, its inputs are the outputs of previous operations, and its results are typically cast back to BF16 before being passed to the next layer (for example, when storing in the KV cache or as input to an activation function). As a result, our experimental results show that significant non-determinism remains on BF16 models.
>
> ### [W3 & Q2] `What are the implications for training?` — Our findings are specific to inference; training is a fundamentally different scenario that actually benefits from BF16's characteristics.
>
> Thank you for raising this important distinction between training and inference. Our analysis focuses specifically on inference reproducibility, and we do not claim that FP32 is superior for training. In fact, there are several key reasons why training is fundamentally different from inference in terms of numerical precision requirements:
>
> 1. **Different Objectives:** During inference, deterministic and reproducible outputs are needed for reliable evaluation and deployment; while training benefits from stochasticity as a form of implicit regularization that helps escape local minima and improve generalization
> 2. **Architectural Differences:** Training only do full prefill computation—the model processes complete sequences without generating new tokens or using KV cache; while during inference, the model generates tokens autoregressively, where each new token depends on all previous tokens, allowing errors to accumulate through the KV cache. Therefore, training do not have the *divergence* problem.
> 3. **Established Best Practices:** Mixed-precision training (FP32 weights, BF16 computation) is the industry standard precisely because it balances: computational efficiency (BF16 operations are faster) and training stability (FP32 master weights).
>
> In summary, our findings about inference reproducibility do not contradict the well-established benefits of mixed-precision training. The core issue we identify—cascading errors through autoregressive generation—is specific to inference and does not manifest in the teacher-forced training paradigm.
>
>
> ### [W4 & Q3] `How is precision handled in key operations like FlashAttention?` — We provide a detailed breakdown and explain why variance accumulates.
>
> Thank you for this suggestion. Here we provide a more detailed discussion on how precision is handled in key Transformer operations and why variance still arises. Let's use a typical BF16 inference setting as an example:
>
> - **Matrix Multiplication**: As discussed in [W2], while the inputs and outputs are typically in BF16, modern Tensor Cores perform the actual accumulation in FP32. However, after the computation, the results are usually cast back to BF16 before being passed to subsequent layers, which reintroduces rounding errors and potential loss of precision.
> - **Layer/RMS Normalization**: These normalization operations require the computation of sums and variances, which are sensitive to numerical errors. For this reason, they are often implemented in FP32 for improved numerical stability, even when the rest of the model uses BF16.
> - **Self-Attention (e.g., FlashAttention)**: Optimized attention kernels such as FlashAttention support both BF16 and FP16. For FP32 models, implementations like xFormers may be used. Regardless of the implementation, self-attention fundamentally relies on floating-point matrix multiplications and reductions to compute attention scores. The order and grouping of these operations can vary depending on GPU architecture, parallelization strategy (e.g., number of heads, sequence length), and specific kernel implementation, making them susceptible to the same non-associativity issues described in Section 2.2. Additionally, intermediate attention scores are often stored in lower precision formats before the final softmax and value-weighting steps, introducing further sources of numerical variation.
>
> In summary, while some components may use higher internal precision, the end-to-end computation involves multiple stages of calculation and data movement where precision can be lost. Our results show that the combination of these small, accumulating errors across the entire model leads to the significant output divergence we observe.
>
>
> ### [Q4] `Could you provide an inverse benchmark to reveal hardware differences?` — A good idea! Here we provide some discussions
>
> Thank you for this excellent suggestion. We believe the reviewer is proposing an "inverse benchmark" where researchers worldwide would run identical model and inference code on their local setups to reveal hardware-induced differences. This is indeed a compelling idea that aligns perfectly with our paper's findings.
>
> In fact, our experimental framework already serves this purpose to some extent. The code we provide (linked in the abstract) allows researchers to reproduce our experiments on their own hardware configurations. Our results in Table 3, showing significant variance across 12 different hardware/system configurations, essentially demonstrate what such an inverse benchmark would reveal.
>
> However, creating a more formalized "inverse benchmark" as the reviewer suggests presents several challenges:
>
> 1. **Attribution Complexity**: As our ablation studies (Section 3.4) show, the variance arises from complex interactions between multiple factors—GPU type, GPU count, batch size, driver versions, and CUDA kernel implementations. Isolating which specific factor causes a particular difference would require extensive controlled experiments.
> 2. **Stochastic Nature**: The hardware-induced differences manifest probabilistically. As shown in Figure 5, not all examples diverge even under BF16, and when they do, the divergence point varies. This makes it difficult to create a simple deterministic test that reliably exposes hardware differences.
> 3. **Standardization Challenges**: The combinatorial explosion of hardware configurations (GPU models × driver versions × CUDA versions × framework versions) makes it impractical to maintain a comprehensive mapping of expected outputs for each configuration.
>
> Despite these challenges, we encourage researchers to use our code to test on their own hardware and contribute their findings, which could collectively build the kind of cross-platform reproducibility map the reviewer envisions. This crowdsourced approach may be more practical than a centralized benchmark given the vast diversity of hardware configurations in use today.
>
>
> ### Regarding Originality
>
> We appreciate the reviewer's feedback on originality and would like to clarify our contributions. While the concepts of floating-point non-associativity and non-determinism in deep learning are known, our work provides the **first systematic and quantitative investigation into how the interplay of hardware configurations (GPU type, tensor parallel size) and system settings (batch size) specifically impacts the reproducibility of modern LLM reasoning.**
>
> Prior to our work, performance discrepancies were often anecdotally observed but not rigorously analyzed. Our contributions are:
> 1.  We move beyond anecdotal evidence to empirically demonstrate and quantify the problem across multiple models and benchmarks, showing that BF16 can lead to accuracy variations of up to 9% (Table 3).
> 2.  We isolate the root cause in the context of LLMs: the interaction between the non-associativity of floating-point math and the minimal probability gaps between competing tokens in reasoning tasks (Figure 3 and 4).
> 3.  We propose LayerCast, a practical and lightweight solution that addresses this problem by achieving FP32-level determinism with a memory footprint much closer to 16-bit models.
>
> We believe this comprehensive analysis, which connects low-level hardware characteristics to high-level model behavior and provides a practical solution, represents a significant and original contribution to making LLM evaluation more reliable and scientific.

---

> > ### Comment · Reviewer_EEhu · 2025-08-01
> > **Analysis of the root cause of difference.**
> >
> > I acknowledge all your detailed answers.
> >
> > Regarding Q1: What I would have liked to see is really an analysis of maybe the model implementations you tested on, whether the difference comes from additions in bf16 vs. fp32 (and non-associativity, which you have for both but is more impactful for bf16), or if the difference is in rounding down to bf16 and up again for the computation in the next layer.
> > Non-associativity of addition is not the single cause of loss of precision in floating point operations, and in my view this could have been analyzed in a bit more detail (e.g. by looking into the very common FlashAttention library for how accumulations are handled). Because if all accumulations are done in FP32 anyways, the main cause be rounding errors instead of non-associativity.
> >
> > Regarding Q4: Of course there would be some attribution complexity. The goal would be to gather as much system information as possible and then later do a data analysis on the observed results. This should then point out configurations that are impactful even beyond precision errors.
> >
> > Despite these short-comings I still recommend the paper for acceptance as it shows the actual impact of lower precision inference and shows a memory efficient solution.

---

> > > ### Author Response · Authors · 2025-08-05
> > >
> > > ### `Non-associativity vs. rounding errors` — Non-associativity is the primary driver of hardware variance
> > >
> > > Thank you for this deeper technical question.
> > >
> > > To clarify: while both non-associativity of floating-point addition and rounding between BF16 and FP32 can introduce numerical errors, our results show that **non-associativity is the main cause of cross-hardware variability**, not rounding. This is because rounding (i.e., casting up to FP32 and back down to BF16) is performed using deterministic rules (just ignore the lower 16-bit mantissa tails or pad zeros on mantissa) and will produce the same results on any hardware setups, given the same input. In contrast, non-associativity means that the order in which additions and other reductions are performed can change the final result, and this order depends on hardware factors like GPU count, batch size, and kernel implementation. **This is why we observe output differences across different hardware setups, even when all use the same precision and rounding logic**. That being said, even with FP32 accumulation (as in FlashAttention), the way operations are grouped and ordered in parallel computation can differ, leading to small but impactful discrepancies.
> > >
> > > We will revise Section 2.2 to make this distinction between non-associativity and rounding more explicit.
> > >
> > >
> > > ### `Community benchmark initiative` — We agree and are exploring implementation paths
> > >
> > > We completely agree this would be a valuable community resource. We are exploring ways to enhance our released code to automatically collect comprehensive system information alongside inference outputs, potentially creating a public repository where researchers can contribute results. Even partial coverage of the vast configuration space would help identify which GPU architectures or driver versions consistently show higher variance. We welcome collaboration on this effort and will mention this future direction in our revised paper.

---

### Official Review · Reviewer_VHiC · 2025-07-01

**Clarity:** 3
**Significance:** 3
**Originality:** 3
**Rating:** 5
**Confidence:** 3

**Summary:**

The paper investigates the importance of numerical precision on the reproducibility of LLM evaluation results. Specifically, it found that inference using the commonly adopted BF16 precision is sensitive to variations in hardware and system configurations, and increasing the number of mantissa bits in the numerical format can mitigate this issue. Hence they proposed an inference pipeline that performances all computation in FP32 while having model weights remain in BF16.

**Questions:**

Please see weaknesses for the questions. Having greater coverage in the experiments as asked would significantly increase the robustness of this study and better support the claims.

**Ethical Concerns:**

["NO or VERY MINOR ethics concerns only"]

**Final Justification:**

The authors have provided additional experimental results that have addressed my concerns regarding the robustness of the results.

**Limitations:**

Please see weaknesses for limitations.

**Paper Formatting Concerns:**

nil

**Quality:**

3

**Strengths And Weaknesses:**

**Strength**
- The paper addresses an important problem of the reliability and reproducibility of LLM benchmarking, especially for reasoning models.
- Specifically, it highlights an often overlooked issue of numerical precision in experimental set-up and how it affects reproducibility.
- The metrics used and analysis in the experiments are reasonable and detailed.
- The proposed solution is simple and easy to implement.

**Weaknesses**
- The main weakness of the paper seems to be in its coverage of experimental settings to validate the claims. The experiments have all been run on fairly similar model sizes (7-8B) and 3 model families, on a fairly limited set of math/coding reasoning tasks. Given the claims of the paper that implies the prevalence of this issue, it is important that the paper has more coverage in its experiments to better support its claims. For example, testing on larger models with higher accuracy and stability in responses may result in much smaller variations due to numerical precision. Similarly for other types of non-math reasoning or long generation tasks.
- It is likely important that the paper also study the impact of the choice of inference backend, i.e., whether the use of vLLM had an impact on the phenomenon that has been observed.

---

> ### Author Rebuttal · Authors · 2025-07-31
>
> ### [W1] `Limited coverage of models and tasks` — We are providing new experiments on larger models and diverse tasks.
>
> Thank you for the suggestion on greater coverage in experiments. Here we provide additional results on a larger model and a new reasoning dataset to strengthen our claims about the prevalence of reproducibility issues.
>
> **Larger Model Evaluation (Qwen3-32B):**
> To demonstrate the impact of numerical precision on reproducibility on larger models, we benchmarked Qwen3-32B on AIME24 dataset under 12 different settings as reported in our paper, which cover 2 GPU counts (2 and 4), 2 GPU version (A100 and L40S), and 3 batch sizes (8, 16, 32). Since Qwen3-32B model cannot be loaded into 2 L40S GPUs with FP32, we have 9 feasible configurations for FP32. The benchmark results are shown below:
>
> |  | 4A100-BS8 | 4A100-BS16 | 4A100-BS32 | 4L40S-BS8 | 4L40S-BS16 | 4L40S-BS32 |
> | ------- | -------- | -------- | -------- | -------- | -------- | -------- |
> | BF16     | 0.8     | 0.8333     | 0.8333     | 0.7333     | 0.8333     | 0.7     |
> | FP32     | 0.7667     | 0.7667     | 0.7667     | 0.7667     | 0.7667     | 0.7667     |
>
> |  | 2A100-BS8 | 2A100-BS16 | 2A100-BS32 | 2L40S-BS8 | 2L40S-BS16 | 2L40S-BS32 |
> | ------- | -------- | -------- | -------- | -------- | -------- | -------- |
> | BF16     | 0.7667     | 0.8333     | 0.7333     | 0.8333     | 0.7667     | 0.7333     |
> | FP32     | 0.7667     | 0.8     | 0.7667     | OOM     | OOM     | OOM     |
>
> |Std@Acc / AIME24|BF16|FP32|
> |----|----|----|
> |Qwen3-32B|5.02%|1.11%|
>
> These results confirm that larger models exhibit the same reproducibility challenges we identified in 7-8B models, with BF16 showing significant variance while FP32 maintains consistency.
>
> **Diverse Task Evaluation (GPQA Diamond):**
> For new tasks other than math/coding, we ran GPQA Diamond benchmark on DeepSeek-R1-Distill-Llama-8B, and the results are shown below.
>
> | Precision | 4A100-BS8 | 4A100-BS16 | 4A100-BS32 | 4L40S-BS8 | 4L40S-BS16 | 4L40S-BS32 |
> | ------- | -------- | -------- | -------- | -------- | -------- | -------- |
> | BF16     | 0.3384     | 0.3838     | 0.4141     | 0.4343     | 0.3838     | 0.4495     |
> | FP32     | 0.4091     | 0.4545     | 0.4343     | 0.3889     | 0.4242     | 0.4495     |
>
> | Precision | 2A100-BS8 | 2A100-BS16 | 2A100-BS32 | 2L40S-BS8 | 2L40S-BS16 | 2L40S-BS32 |
> | ------- | -------- | -------- | -------- | -------- | -------- | -------- |
> | BF16     | 0.4091     | 0.4192     | 0.4141     | 0.3889     | 0.4394     | 0.4192     |
> | FP32     | 0.4091     | 0.4343     | 0.4091     | 0.4293     | 0.4141     | 0.4444     |
>
> |Std@Acc / GPQA-Diamond|BF16|FP32|
> |----|----|----|
> |DeepSeek-R1-Distill-Llama-8B|3.03%|1.96%|
>
> The above results indicate that the reproducibility issue also exists on GPQA Diamond benchmark, supporting the prevalence of this issue. These additional results strengthen our argument that numerical precision is a critical but overlooked factor in LLM evaluation reproducibility across model scales and task types.
>
>
> ### [W2] `Is this a backend-specific issue?` — The issue is fundamental to GPU hardware and observable across backends.
>
> We thank the reviewer for this insightful suggestion. We agree that it is important to ensure our findings are not specific to a single inference backend.
>
> Our paper's central thesis is that the observed non-determinism originates from a fundamental property of modern computing hardware: the non-associative nature of floating-point arithmetic (as detailed in Section 2.2). As we detail in our paper, operations like `(a+b)+c` are not guaranteed to equal `a+(b+c)` due to rounding errors in finite-precision formats like BF16. This issue is not specific to vLLM but is inherent to how parallel computations are performed on GPUs across all major deep learning frameworks.
>
> Indeed, **this phenomenon is observable across different inference backends**. For instance, a similar issue of non-reproducible outputs with greedy decoding has been reported by users of SGLang [1]. Furthermore, the official PyTorch documentation [2] explicitly discusses this source of non-determinism, attributing it to the low-level CUDA kernels.
>
> In conclusion, while our experiments are implemented using vLLM, the root cause we identify and analyze is at a much lower level of the computing stack, related to hardware-level floating-point computation. Our findings are therefore general and applicable to other inference systems that rely on parallel GPU execution.
>
> [1] SGLang Issue 7916
>
> [2] PyTorch Docs: Randomness

---

> > ### Comment · Reviewer_VHiC · 2025-08-01
> >
> > Thanks for the detailed responses. The additional experiments are useful in strengthening support for the claims re prevalence of the issue across different models and the benchmarks.
> >
> > It is still not clear whether the specific observed empirical phenomenon (e.g. extent of performance gap in experiments) is definitely independent of vLLM, and not an experimental artifact. A smaller-scale, simple experiment demonstrating similar results for inference without vLLM (as compared to inference with vLLM) would be useful in providing some indication.

---

> > > ### Author Response · Authors · 2025-08-05
> > >
> > > ### `Additional verification without vLLM` — HuggingFace experiments confirm the issue is not backend-specific
> > >
> > > Thanks for the suggestion to add experiments without vLLM. Acting on this hint, we performed new runs with the **standard HuggingFace Transformers** implementation — specifically, without vLLM or PagedAttention. The experiment setup is identical to the one in our main paper — 2 GPU counts (2 and 4), 2 GPU version (A100 and L40S), and 3 batch sizes (8, 16, 32), covering 12 settings for each precision.
> > >
> > > We run the experiments on two models on AIME'24:
> > >
> > > | Model/Dataset | Std@Acc BF16 | Std@Acc FP32 |
> > > | -------- | :--------: | :--------: |
> > > | Qwen-2.5-7B-Instruct / AIME’24  | 2.23% | 0  |
> > > | Llama-3.1-8B-Instruct / AIME’24  | 1.51% | 0  |
> > >
> > > Again, the same pattern is shown in the above results — FP32 inference is fully reproducible while BF16 witnesses similar variance to what we reported with vLLM. This demonstrates that the phenomenon is fundamental to GPU hardware and observable across backends.

---

> > > > ### Comment · Reviewer_VHiC · 2025-08-07
> > > >
> > > > Thanks for the additional results. My concerns are addressed, and I raise my score.

---

### Note · Authors · 2025-08-11

We would like to thank reviewers and AC for their efforts. We are glad that the reviewers acknowledged the strengths of our work, specifically:

- Studies the important and often-overlooked problem of numerical precision's impact on the reproducibility of LLM evaluation (Reviewers VHiC, EEhu, mWEn, 3spC).
- Provides a systematic, thorough, and convincing experimental analysis to demonstrate the significance of this issue (Reviewers VHiC, mWEn, 3spC).
- Proposes LayerCast, a simple and effective solution to mitigate reproducibility issues without significant memory or performance overhead (Reviewers VHiC, EEhu, mWEn).
- Highlights a problem with the potential for long-lasting impact on the field (Reviewer mWEn).

During the rebuttal period, we provided more experiments and discussions to address common questions from the reviewers. Specifically, we:

- Expanded our experimental coverage by running new tests on larger models (Qwen3-32B) and more diverse, non-mathematical reasoning tasks (GPQA-Diamond) to demonstrate the prevalence of the reproducibility issue.
- Verified that the issue is not backend-specific by conducting new experiments using the standard HuggingFace Transformers library, confirming that the phenomenon is fundamental to the hardware and not an artifact of a specific inference engine like vLLM.
- Provided a deeper technical analysis of how precision is handled in key Transformer operations like FlashAttention and offered a detailed hypothesis on why different setups exhibit different levels of variability.
- Strengthened the evaluation of our proposed solution, LayerCast, by providing new results on additional models to confirm its effectiveness and general applicability.

---

### Decision · Program_Chairs · 2025-09-17

**Decision:**

Accept (oral)

**Comment:**

The paper shows that choice of floating point precision can lead to significant changes in inference accuracy reported for reasoning models.

Strengths: Extensive study showing the extent of variation, important to the interpretation of reasoning results, provide a solution to address reproducibility issues without incurring memory overhead
Weaknesses: one of those papers where in hindsight, it seems obvious that FP precision should impact results

During the rebuttal, authors strengthened their argument by experiments on models of varying sizes and a new task beyond math and code. All reviewers appreciated the insight from this paper and the extensive experiments to support it. I think it is a timely paper and worthy of broader discussion.